# Probing atomic physics at ultrahigh pressure using laser-driven implosions

S. X. Hu [1,2] ✉, David T. Bishel [1,3], David A. Chin [1,3], Philip M. Nilson [1] ✉, Valentin V. Karasiev [1], Igor E. Golovkin [4], Ming Gu[4], Stephanie B. Hansen[5], Deyan I. Mihaylov [1], Nathaniel R. Shaffer[1], Shuai Zhang [1] & Timothy Walton[4]

Spectroscopic measurements of dense plasmas at billions of atmospheres provide tests to our fundamental understanding of how matter behaves at extreme conditions. Developing reliable atomic physics models at these conditions, benchmarked by experimental data, is crucial to an improved understanding of radiation transport in both stars and inertial fusion targets. However, detailed spectroscopic measurements at these conditions are rare, and traditional collisional-radiative equilibrium models, based on isolated-atom calculations and ad hoc continuum lowering models, have proved questionable at and beyond solid density. Here we report time-integrated and time-resolved x-ray spectroscopy measurements at several billion atmospheres using laser-driven implosions of Cu-doped targets. We use the imploding shell and its hot core at stagnation to probe the spectral changes of Cu-doped witness layer. These measurements indicate the necessity and viability of modeling dense plasmas with self-consistent methods like density-functional theory, which impact the accuracy of radiation transport simulations used to describe stellar evolution and the design of inertial fusion targets.

The physics of warm and hot dense matter can unravel the mysterious inner workings of planetary cores and stellar interiors[1]. These conditions span a large range of densities and temperatures ($\rho = 10^0$–$10^6$ g cm$^{-3}$ and $T = 10^3$–$10^7$ K), with pressures varying from ~1 Mbar (or, one million times that of Earth's atmospheric pressure; 1 Mbar = $10^{11}$ Pa) to ~500 Gbar (1 gigabar = $10^{14}$ Pa). Understanding the physics of matter at such ultrahigh pressures can have many applications, including determining the age of the Universe through white dwarf cosmochronometry[2], interpreting astrophysical observations[3–5], and designing high-performance inertial fusion targets[6–8]. Thanks to technological advances in high-power lasers (including x-ray free electron lasers) and pulsed-power machines, this extreme state of matter can now be accessed in the laboratory[9–11], but only for a short period of time (picosecond to microsecond timescales) depending on the driver and experimental geometry. Nonetheless, these techniques provide a unique "window" for interrogating the physics of matter at extreme conditions. The implosion spectroscopy measurements and model development presented in this work aim to reveal a more-detailed picture of atomic physics in dense-plasma environments at billion atmosphere (Gbar) pressures. Spherically-convergent techniques uniquely access the gigabar pressure regime in experiments, providing the necessary data to test atomic physics models for warm and hot dense plasmas.

X-ray spectroscopy, a common and sometimes only means to diagnose and understand short-lived plasmas, measures x-ray emission and absorption with spatial, spectral, and/or temporal resolution[12–16]. Observing atomic line positions and spectral widths can reveal the physical processes that are occurring inside the system. Reliable atomic and plasma physics models are required to interpret these spectral signatures and have generally proven to be adequate for

[1]Laboratory for Laser Energetics, University of Rochester, 250 East River Road, Rochester, NY 14623-1299, USA. [2]Department of Mechanical Engineering, University of Rochester, Rochester, NY 14623, USA. [3]Department of Physics and Astronomy, University of Rochester, Rochester, NY 14627, USA. [4]Prism Computational Sciences, 455 Science Drive, Madison, WI 53711, USA. [5]Sandia National Laboratories, 1515 Eubank SE, Albuquerque, NM 87185-1196, USA. ✉ e-mail: shu@lle.rochester.edu; pnil@lle.rochester.edu

spectroscopically diagnosing classical/ideal plasmas[17–20]. In this regime, collisional-radiative equilibrium (CRE) models[21,22] are successfully used, which combine accurate atomic data from isolated atom calculations with appropriate continuum-lowering models to describe dilute plasma effects (e.g., ionization, screening, and broadening). This approach can provide guidance, for example, on the inference of plasma density and temperature[17–20]. However, with increasing energy density, experimental measurements over the last decade have revealed potential inconsistencies with traditional CRE treatments.

For instance, experimental measurements[23,24] on the K-edge shift of solid-density aluminum plasmas (heated by x-ray free electron lasers) favored the continuum lowering model developed by Ecker and Kroll[25], while shock-compression experiments[26] on the same material gave better agreement with a different continuum-lowering model by Stewart and Pyatt[27]. In addition, iron opacity measurements[28] at pressures below 1 Mbar showed very good agreement with traditional CRE-type opacity calculations, while significant disagreements[29,30] were found between measurements and theory at elevated densities and temperatures (for example, at around 10 Mbar for iron plasmas). It remains an "unsolved mystery" to this day, even though much effort has been applied to this open question from both theoretical and experimental perspectives[30–32].

Today, one can accurately compute the electronic energy levels of an isolated atom by solving the many-body Schrödinger or Dirac equations, for which the calculation precision can be improved systematically by varying the sophistication of the methods that are implemented, from the simplest Hartree–Fock method to advanced multi-configuration interactions. However, when atoms are put into a non-ideal (i.e., strongly-coupled and/or degenerate) plasma environment, significant discrepancies appear between detailed spectroscopic measurements and calculations. One outstanding example is the inconsistency of hydrogen line broadening in the dilute, but cold ($n_e = 10^{15}$–$10^{18}$ cm$^{-3}$ and T = $10^3$–$10^5$ K) photospheric plasmas of white dwarfs[33], in which plasma conditions inferred from the broadening of different lines in the same plasma can vary significantly, even amongst the best atomic physics models that are currently available. These variations can have significant implications for deducing the mass and age of white dwarfs by affecting the standard candle for cosmochronometry[2]. A similar situation occurs in warm dense plasmas under high-energy-density (HED) conditions, in which high-density effects (many-body coupling) and quantum electron degeneracy can drastically alter atomic physics relative to the isolated case. Reconciling how atomic physics changes in such non-ideal plasmas demands progress in both experiments and theory, which must account for the plasma environment self-consistently.

Over the last few years, high-resolution absorption and fluorescence spectra have been used in magnetically driven inertial fusion (cylindrical liner) experiments to study the electronic structure of warm dense matter under extreme compression[16,34]. These studies have shown that a self-consistent field model based on density-functional theory (DFT) could reproduce K-edge and fluorescence line shifts at independently diagnosed, imploded plasma conditions (10 eV and $n_e = 10^{24}$ cm$^{-3}$), but collisional-radiative models with ad-hoc density effects could not reproduce the measured x-ray spectra[34]. A pure compressional experiment without thermal or ionization effects measured density-induced shifts in the K$_\beta$ line of cobalt at 8 Mbar, in good agreement with a self-consistent DFT model, and found significant differences among the predictions of several CRE models[35]. It is also noted that DFT-based modeling has been successfully applied to x-ray near-edge absorption spectroscopy (XANES) for warm-dense matter[36–38]. These earlier XANES experiments showed absorption features in good agreement with DFT calculations[36–38]. Extension of these studies to gigabar pressures are very important because of their relevance to fundamental dense plasma theory, inertial fusion energy, and laboratory astrophysics.

Here, we report x-ray spectroscopy measurements at gigabar pressures using laser-driven implosions. These measurements are used to test a DFT-based multi-band kinetic model (VERITAS), which is developed in this work. The VERITAS model uses DFT-derived band (atomic level) information to compute the radiative transition rates that can be coupled to the radiation transfer equation to describe the radiation generation and transport processes in a dense plasma. With Cu (as a witness element) doped inside a 30-μm-thick plastic shell implosion, we performed time-integrated and time-resolved Cu K$_\alpha$ emission (the 2p→1s transition) and 1s-2p absorption measurements during shell stagnation. Both of these inverse processes are observed on the same experiment; photo-ionization of 1s electrons enables K$_\alpha$-emission, and thermal-ionization of 2p electrons enables 1s-2p absorption. These observations are directly connected to the time-dependent atomic ionization balance in the assembled dense plasma. The system is further constrained by integrated measurements of the compressed areal density (ρR), neutron yield and bang-time, and ion temperature, allowing the spectroscopic data to differentiate the DFT-based kinetic model from traditional treatments based on isolated-atom calculations and ad hoc continuum-lowering models.

The paper is organized as follows: first, the necessity of a reliable atomic physics model for interpreting x-ray spectroscopic measurements is demonstrated using a surrogate dense-plasma object. The experimental results are then presented with a detailed spectral comparison between measurements and simulations based on traditional atomic physics models and the DFT-based approach that is developed in this work. Finally, the implications of these results for understanding dense plasma environments are discussed.

## Results

### Surrogate dense-plasma object

To illustrate why a reliable atomic physics model is required a priori for interpreting dense plasma spectroscopy measurements, we construct a surrogate dense-plasma object in spherical geometry and compute synthetic x-ray spectra based on different atomic physics treatments. The surrogate plasma object consists of a 20-μm-radius core of 1%-Ar– doped deuterium plasma, having a given mass density of $\rho = 10$ g cm$^{-3}$ and temperature of $kT = 1000$ eV, surrounded by four concentric homogenous shells of CH or Cu-doped CH with densities, temperatures, and thicknesses shown in Fig. 1a. The Cu-doped CH plasma serves as a "witness" layer (denoted as CHCu[2%]), which has 2% atomic fraction of Cu, uniformly mixed into the CH plasma.

Synthetic spectra, calculated with three different atomic physics models, are shown in Fig. 1b, c for the same CHCu[2%] density of $\rho = 20$ g cm$^{-3}$, but different temperatures ($kT = 200$ eV and $kT = 300$ eV, respectively). The traditional CRE simulations used an atomic database (ATBASE), implemented by the Spect3D software package, based on a kinetic description for the atomic level populations, by which levels are populated and depopulated by radiative and collisional processes, and coupled to nonlocal radiation transport. As discussed above, these CRE models need to invoke continuum-lowering models to "destroy" bound levels and account for plasma effects (pressure ionization and lowering of ionization thresholds). The remaining results in Fig. 1b, c come from VERITAS, a new DFT-based multi-band kinetic model for dense plasma spectroscopy.

### The VERITAS code

The details of VERITAS can be found in Methods; here we briefly describe its essential components: (1) the electronic structure of dense Cu-doped CH plasma is determined self-consistently by DFT through quantum molecular-dynamics simulations using all-electron potentials, for a given density and temperature grid; (2) certain electronic bands, such as the 1s, 2p, 3p, and continuum, are chosen to be included in the model – the oscillator strengths among these bands are calculated for the considered radiative dipole transitions; and (3) the kinetic

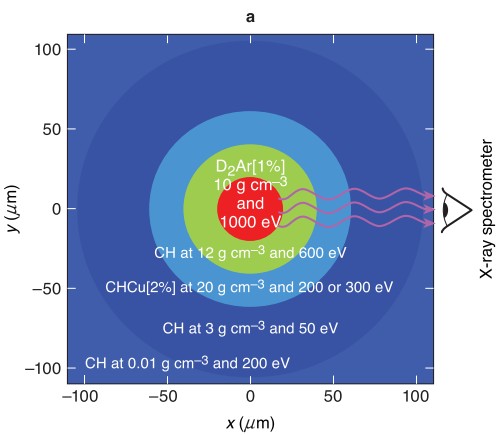

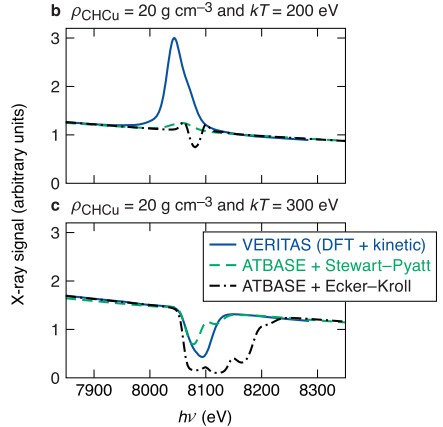

**Fig. 1 | Illustration of predicted spectroscopic differences for warm-/hot-dense plasmas by different atomic physics models. a** Schematic of a surrogate dense-plasma object consisting of a Cu-doped CH plasma layer for spectroscopy. **b** The predicted Kα emission signal from the doped Cu layer of mass density of $\rho = 20\,\mathrm{g\,cm^{-3}}$ and kT = 200 eV, by three different models: *VERITAS* (blue solid),

collision radiative equilibrium (*CRE*) models with continuum lowering of Stewart–Pyatt (green long dash) and Ecker–Kroll (black dash-dotted). **c** The predicted *1s–2p* absorption feature from the doped Cu layer at mass density of $\rho = 20\,\mathrm{g\,cm^{-3}}$ and temperature kT = 300 eV.

equation invoking the DFT-determined transition rates is used to describe the population change in these energy bands due to radiative transitions, which is coupled to the non-local radiation transport equation to ensure that the local radiation field is consistent with the band population. In contrast to a traditional *CRE* treatment, our DFT-based kinetic code – *VERITAS*—explicitly accounts for the micro-physical interactions among ions and the dense plasma environment. Energy band shifting and ionization balance are self-consistently described, without invocation of an ad hoc continuum lowering model. This model development is based on the preliminary success of treating warm-dense plasmas as quantum many-body systems[39–44] with mean-field DFT.

The *VERITAS* predictions for the surrogate dense-plasma object prescribed by Fig. 1a are indicated by the blue solid curves in Fig. 1b, c. For the case of kT = 200 eV, Fig. 1b shows that when the hot-spot radiation streams out through the CHCu[2%] layer, high-energy photons excite or ionize the *1s* core electron of Cu, leading to Kα emission (due to the *2p→1s* transition). As the temperature of the Cu-doped layer increases to kT = 300 eV, the spectroscopic feature changes from Kα emission to the dominant *1s-2p* absorption, shown by Fig. 1c. This feature change is caused by the appreciable depletion of the Cu *2p* population at this higher temperature.

Compared to the DFT-based *VERITAS* model, the two *CRE* models marked as "ATBASE + Stewart–Pyatt" (green dashed curve) and "ATBASE + Ecker–Kroll" (black dash-dotted curve) give quite different spectroscopic predictions for the same plasma conditions. These differences are quantitatively distinguishable: (1) the Kα-emission peak shifts by ~20 eV in the two *CRE* models when compared to *VERITAS* for the low temperature case shown in Fig. 1b; (2) the Kα-emission peak from *VERITAS* is more than two-fold stronger than both *CRE* models, while the Ecker–Kroll model predicts a *1s-2p* absorption feature even at kT = 200 eV; (3) at a higher temperature of kT = 300 eV, all models predict the *1s-2p* absorption although the ATBASE + Ecker–Kroll model gives a wider and stronger absorption feature as indicated by Fig. 1c; and (4) at this temperature the *VERITAS* and Stewart–Pyatt models give a similar absorption width, but the latter shows a "double-dip" feature.

To investigate what detailed atomic physics drives these different observations, we compare in Table 1 the free-electron density, average Cu ionization ($Z_{Cu}^*$), and Cu *2p* population predicted by the following spectral models: ATBASE + Stewart–Pyatt (*Spect3D* - a *CRE* code), DFT + QMD (*VERITAS*), DFT + AA (*Muze*), FAC + AA [flexible atomic code with plasma environment inferred by an average-atom (AA) model] and one other *CRE* code – SCRAM. The "FAC + AA" model uses FAC-code calculations for the atomic structure of a Cu atom that is "embedded" in a CH plasma mixture in which the plasma environment is described by an average-atom (AA)-type model. It embodies a similar "*spirit*" to DFT with a self-consistent-field (SCF) calculation of plasma screening for an atom embedded in a plasma mixture.

The comparison indicates that both $Z_{Cu}^*$ (which governs Kα shifts) and the depletion of 2p (which controls the Kα intensity) are similar among the three DFT-based models [*VERITAS*, *Muze*, and *FAC + AA*]. By contrast, the traditional CRE models with similar ad-hoc continuum-lowering treatments differ from the self-consistent models and even from each other. These noticeable differences have motivated us to design and perform experiments in a similar regime, aiming to inform the development of a more-reliable HED atomic physics model for radiation generation and transport in dense plasmas.

### Experimental setup and diagnostics

The experiment used a spherical, laser-driven implosion on the Omega Laser Facility. The target, shown schematically in Fig. 2a, consists of a 30-μm–thick polystyrene (CH) shell with a 10-μm–thick layer uniformly doped with 2% Cu (atomic fraction) and a 1%-Ar-doped deuterium (D2Ar[1%]) core fill. The 10-*μm*-thick Cu-doped layer begins ~3-

### Table 1 | Comparisons of free-electron density ($n_e$), average ionization $Z_{Cu}^*$ and 2p-population ($f_{2p}$) of Cu in warm-/hot-dense plasmas

| Models | kT = 200 eV | | | kT = 300 eV | | |
|---|---|---|---|---|---|---|
| | $n_e$ (cm⁻³) | $Z_{Cu}^*$ | $f_{2p}$ | $n_e$ (cm⁻³) | $Z_{Cu}^*$ | $f_{2p}$ |
| ATBASE + Stewart–Pyatt (*Spect3D*) | $5.1 \times 10^{24}$ | 13.76 | 6.00 | $5.5 \times 10^{24}$ | 15.78 | 5.93 |
| SCRAM + ion-sphere | $5.1 \times 10^{24}$ | 14.39 | 5.84 | $5.5 \times 10^{24}$ | 16.37 | 5.25 |
| DFT + QMD (*VERITAS*) | $5.0 \times 10^{24}$ | 12.88 | 5.87 | $5.3 \times 10^{24}$ | 14.96 | 5.27 |
| DFT + AA (*Muze*) | $5.0 \times 10^{24}$ | 12.52 | 5.87 | $5.5 \times 10^{24}$ | 14.43 | 5.24 |
| FAC + AA (*FAC*) | $4.2 \times 10^{24}$ | 12.86 | 5.87 | $4.9 \times 10^{24}$ | 15.19 | 5.19 |

The CHCu[2%] plasma mixture has a mass density of ρ = 20 g cm⁻³ and two different temperatures. These results are predicted from different atomic physics models. Comparisons of these predicted physical quantities demonstrate the differences between two categories of atomic physics models for warm- or hot-dense plasmas: DFT-based models *vs.* traditional collisional-radiative models. The quoted value of $Z_{Cu}^*$ for *VERITAS* was calculated from the Thomas–Fermi average-atom model to provide a comparison to the other models, even though DFT calculations do not need to define $Z_{Cu}^*$. Additional details of these models can be found in the Methods section.

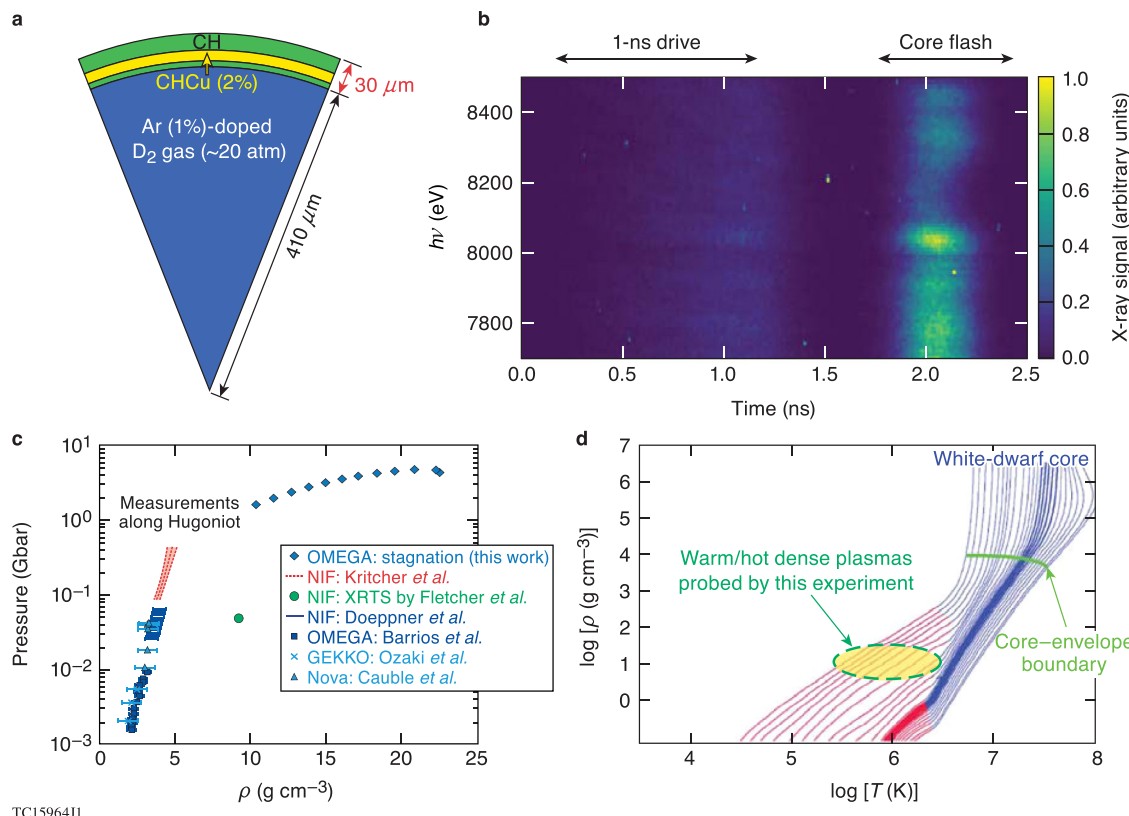

**Fig. 2 | Time-resolved x-ray spectroscopy experiment of warm-/hot-dense plasmas at white-dwarf's envelope conditions of Gbar pressures. a** Schematic targets for implosion spectroscopy on OMEGA. **b** Example of streaked spectra measured in experiments. **c** The pressure-density region probed by various HED experiments: GEKKO[47], OMEGA[48], Nova[45], NIF by Doeppner et al.[11], NIF by Kritcher et al.[51,52], NIF by Fletcher et al.[49], as well as non-Hugoniot work by Doeppner et al.[50] on NIF. **d** The density-temperature conditions of a typical white dwarf of 0.6M⊙

(0.6 solar mass) as it's cooling down from hot and young state (right) to older and colder structures (left). Convective regions in the stars are shown in red. The regime probed by the experiments is shown by the green dashed circle. Inferred from *DRACO* simulations, the plasma temperature and density conditions of the imploding Cu-doped layer vary from $kT \approx 10–50$ eV and $\rho \approx 2–10$ g cm$^{-3}$ (in-flight stage) to $kT \approx 200–500$ eV and $\rho \approx 10–25$ g cm$^{-3}$ during the stagnation.

μm from the inner surface of the CH shell. The target was imploded by 60 laser beams on OMEGA with a 1-ns square pulse having a total energy of ~26 kJ. When the laser pulse irradiates the spherical capsule, laser ablation launches a strong shock wave that compresses the target. After the shock breaks out of the inner surface of the shell into the gas-filled core, the shell is accelerated inwards until it stagnates at a certain radius. At stagnation, the contained gas is compressed and heated to form a hot core, which emits x-rays that probe the stagnating shell and enable our spectroscopic measurements.

Both time-integrated and time-resolved x-ray spectrometers were used to record the emergent radiation spectrum (see Methods for further details). Figure 2b shows a typical time-resolved x-ray spectrum in the photon energy range 7800 to 8600 eV, which clearly indicates the Cu emission and absorption features of interest during the shell stagnation and core flash.

## Table 2 | Comparisons of implosion performance between experiment and *DRACO* simulation

| Shot# 97628 | Experiment | *DRACO* Simulation |
|---|---|---|
| D-D Neutron Yield | $(7.3 \pm 0.4) \times 10^9$ | $6.2 \times 10^9$ |
| $\langle T_i \rangle_n$ (keV) | $2.2 \pm 0.5$ | 2.6 |
| $\langle \rho R \rangle_n$ (mg cm$^{-2}$) | $67 \pm 7$ | 66.5 |
| Neutron bang time (ns) | $2.04 \pm 0.05$ | 2.08 |

The experimental measurements of neutron yield, neutron-averaged ion temperature, neutron-averaged areal density, and neutron bang time are compared with radiation-hydrodynamic simulations by *DRACO*.

### Implosion performance

These high-adiabat ($\alpha = 10$) and low-velocity (~250 km s$^{-1}$) implosions are stable to laser imprint and other perturbations, as indicated by one- and two-dimensional radiation-hydrodynamic simulations using the *LILAC* and *DRACO* codes (see below), as well as integrated experimental measurements of the implosion performance (see Table 2). Table 2 shows that the DD fusion neutron yield, neutron-averaged ion temperature $\langle T_i \rangle_n$, neutron-averaged shell areal density $\langle \rho R \rangle_n$, and neutron bang-time are in close agreement with *LILAC* and *DRACO* simulations. Based on these observations, we can reasonably process the radiation-hydrodynamic-simulations with atomic physics models to obtain synthetic x-ray spectra and compare them to experimental measurements.

Compared to other shocked-CH studies[45–52] carried out mainly along the principal Hugoniot, our experiment has extended the pressure and density conditions at which both time-integrated and time-resolved x-ray spectroscopic measurements have been conducted: gigabar (Gbar) pressures and ~15–20 × solid-density, as indicated by Fig. 2c. Inferred from DRACO simulations, the plasma temperature and density conditions in the imploding Cu-doped layer vary from $kT \approx 10–50$ eV and $\rho \approx 2–10$ g cm$^{-3}$ (in-flight stage) to $kT \approx 200–500$ eV and $\rho \approx 10–25$ g cm$^{-3}$ during stagnation. The corresponding pressure in the compressed shell changes from ~50 Mbar to a maximum value approaching ~5 Gbar. When one casts these dense plasma conditions achieved on OMEGA to the density-temperature conditions of a typical white dwarf (0.6 M⊙) during its cooling phase, Fig. 2d shows the experiment can potentially probe the equation of state and transport properties of the convective region of a white dwarf's envelope.

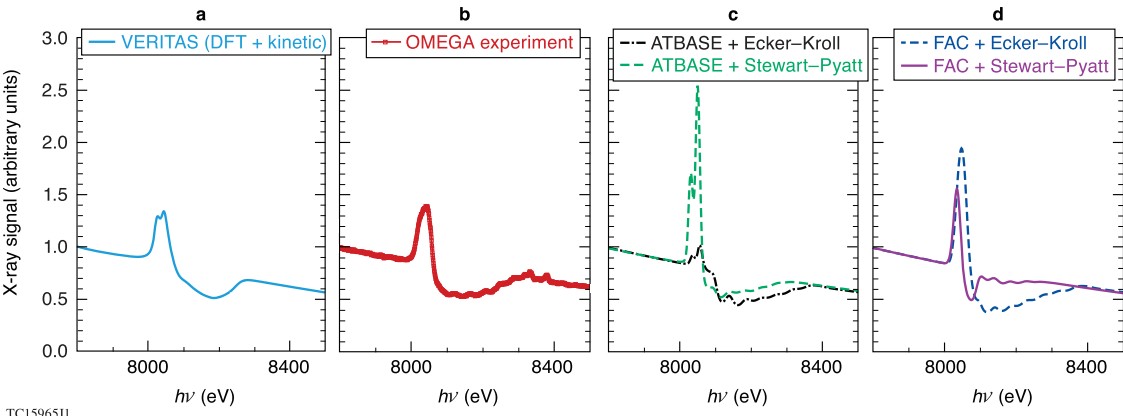

**Fig. 3 | Comparison of time-integrated K$_\alpha$-emission and *1s-2p* absorption signals between experiment and models. a** The DFT-based *VERITAS* calculations. **b** The experimental measurement. **c** The *CRE* model calculations with atomic database (ATBASE) in combination with Stewart–Pyatt and Ecker–Kroll continuum lowering models. **d** The *CRE* model calculations using flexible atomic physics (FAC) code calculation with the two continuum lowering models. The time integration in calculations has been done from $t = 1.7$ ns to $t = 2.4$ ns during the hot-spot flash, with snapshots for each 20-ps time interval.

Accurate knowledge about ionization balance in such conditions could directly affect the modeling of conduction and the radiative cooling of white dwarfs.

## Spectroscopic modeling

Using the *DRACO*-simulated dynamic plasma conditions, we investigated x-ray generation and transport through the target using two *CRE* models (ATBASE and FAC) and the DFT-based kinetic code *VERITAS*. The predicted time-integrated spectra are compared with the experimental measurements in Fig. 3, in which the x-ray signal is plotted as a function of photon energy [all normalized to the continuum signal level at 7800 eV]. The experimental spectra (Fig. 3b) show both the pronounced K$_\alpha$-emission peaked at ~8042 eV and the *1s-2p* absorption of Cu in the higher photon energy range of 8100–8250 eV. Both the location and amplitude of the emission and absorption features are appropriately captured by *VERITAS* (Fig. 3a).

Figure 3c, d show the *Spect3D* simulation results, in which either the atomic database (ATBASE) or the flexible atomic code (FAC) calculations are combined with the Ecker–Kroll and Stewart–Pyatt continuum lowering models. When these *CRE* results are compared to experiments, they give a conflicting conclusion about the continuum lowering model. Namely, the experimental emission and absorption features are qualitatively reproduced by the two *CRE* simulations of "ATBASE + Stewart–Pyatt" and "FAC + Ecker–Kroll" in Fig. 3d, e (though the emission peaks are too high), while the other two combinations drastically disagree with experiments. This illustrates again the *dilemma* of the traditional spectroscopic treatment for warm dense plasmas: which ad hoc continuum lowering model works better depends on the atomic physics model that is invoked. The resemblance between the FAC + Ecker–Kroll model (Fig. 3d) and experiments is likely coincidental, as other recent measurements[53] of ionization-potential-depression have defied the Ecker–Kroll model.

Overall, the DFT-based *VERITAS* model, without invocation of an ad hoc continuum lowering model, better resembles the observed x-ray signal in the experiments. Nonetheless, one can see that the *VERITAS*-predicted continuum slope, the K$_\alpha$-emission amplitude, and the *1s-2p* absorption width are still slightly mismatched with respect to the experiment. This small spectroscopic discrepancy might be attributed to some unavoidable three-dimensional effects, even though the time-integrated implosion measurements overall agree with 2D *DRACO* simulations. For instance, the stalk perturbation could inject a small and localized portion of the Cu-doped layer closer to the hot spot, which, to some extent, could contribute to the measured

spectra in ways that are not accounted for in the 2D model. These small differences are further discussed in Supplementary Information.

## Time-resolved x-ray spectrum

Experimental and synthetic time-resolved x-ray signals are presented in Fig. 4. The top three panels compare the measured streak-camera image of the x-ray emission and absorption over the core flash (Fig. 4b) with predictions from the ATBASE + Stewart–Pyatt model (Fig. 4a) and *VERITAS* (Fig. 4c). Figure 4d–f give quantitative comparisons at $t = 1.95$ ns, $t = 2.05$ ns, and $t = 2.15$ ns. All these cases are normalized to each other with the same continuum signal level. The experimental time-resolved spectrum shows pronounced K$_\alpha$ emission early in time (Fig. 4d), which changes to a dominant *1s-2p* absorption as time proceeds.

At early times, the DFT-based kinetic model (*VERITAS*) agrees well with the K$_\alpha$ emission measurements, while the ATBASE + Stewart–Pyatt model over-predicts the emission peak and resolves the K$_{\alpha 1}$ and K$_{\alpha 2}$ spectral lines (due to less broadening), as shown in Fig. 4a. At t = 2.05 ns the heat wave reaches the Cu-doped layer, leading to a *1s-2p* absorption "dip" in Fig. 4b. Again, the ATBASE + Stewart–Pyatt model gives a stronger absorption dip at lower photon energy in comparison to experiment, while *VERITAS* shows the same level of *1s-2p* absorption depth.

It is noted that the experimental *1s-2p* absorption feature is somewhat wider than in the model predictions. This slight discrepancy might come from the possibility that regions of the Cu-doped CH were driven deeper towards the hot spot by the stalk or other 3D perturbations in the experiments. Finally, when the shock and heat-wave have propagated through most of the CHCu[2%] layer, *VERITAS* still catches the experimental absorption level and width appropriately (Fig. 4f), while the ATBASE + Stewart–Pyatt model gives somewhat stronger absorption (green dashed curve).

## Discussion

The spectroscopic evolution from K$_\alpha$ emission to *1s-2p* absorption is directly related to the plasma conditions that are dynamically changing in the Cu-doped layer during stagnation. The density and temperature contours of the imploding shell at stagnation are respectively depicted in the upper and lower panels of Fig. 5a, as predicted by *DRACO* simulations. It shows the formation of a hot D2Ar[1%] core, with a return shock reaching the Cu-doped-CH layer, and a heat wave from the hot core propagating outward by thermal conduction.

To further illustrate this process, we plot in Fig. 5b the angularly-averaged plasma density and temperature at the inner and outer

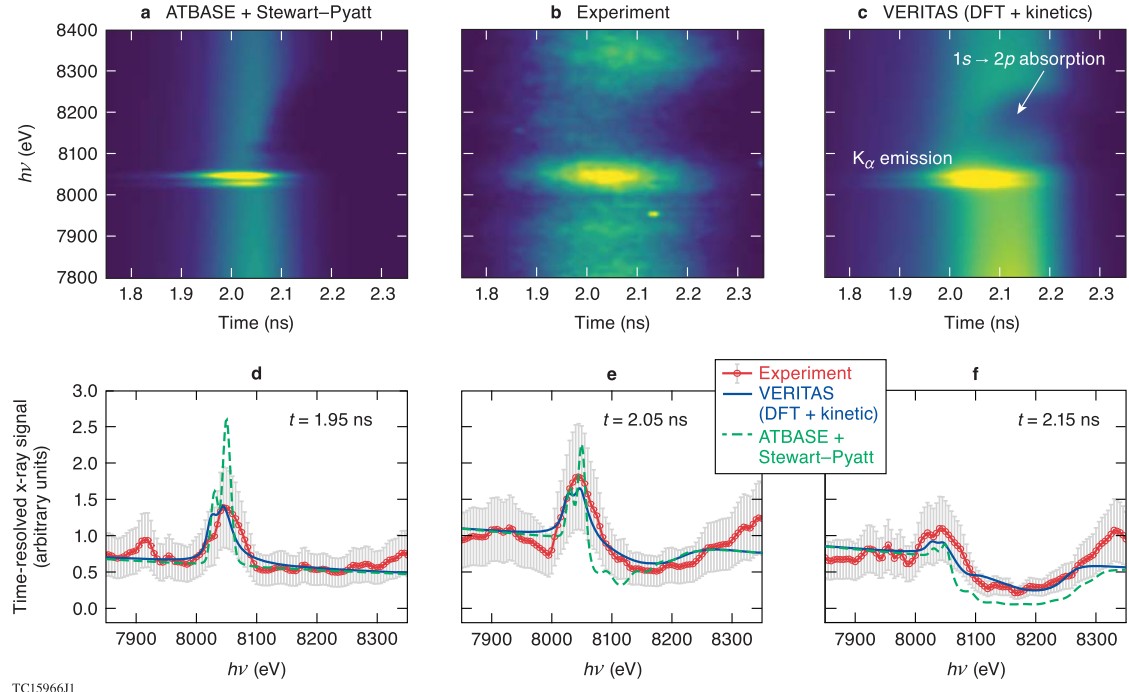

**Fig. 4 | Comparison of time-resolved x-ray signals between experiment and models during the core flash. a** The streaked spectra predicted by traditional *CRE* model (*Spect3D*) with isolated atomic database plus continuum-lowering (Stewart–Pyatt). **b** The experimental measurement. **c** The streaked spectra predicted by *VERITAS* (a DFT-based kinetic model). **d–f** The spectral comparisons among the three cases at three distinct time line-outs: $t = 1.95$-ns, 2.05-ns, and $t = 2.15$-ns. The experimental error bar of ±40% is mainly from x-ray photon statistics of the streaked signal.

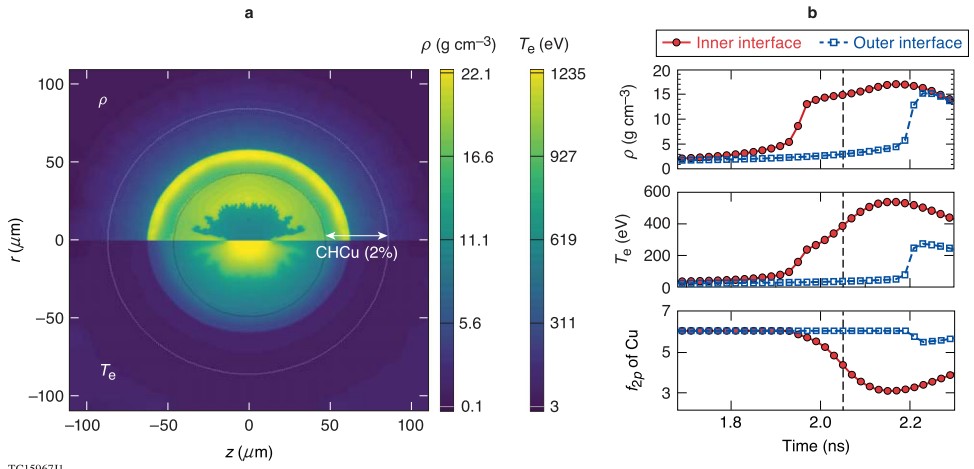

**Fig. 5 | The rad-hydro-predicted warm-/hot-dense plasma conditions during core flash of Cu-doped CH target implosions on OMEGA. a** The density (upper) and temperature (down) contour plots of dense plasma conditions at stagnation ($t = 2.05$ ns) from 2D *DRACO* radiation-hydrodynamic simulation. The inner and outer circles of dotted lines indicate the inner and outer boundary of the Cu-doped layer whose region is marked by the black arrow. **b** The time evolution of plasma ρ/T conditions as well as the population of Cu's 2p state in the Cu-doping region (inferred by *VERITAS*) during the core flash, in which red symbols represent the situation at the inner interface and blue symbols are for the outer interface. The stagnation time ($t = 2.05$ ns) is marked by the vertical dashed line.

surfaces of the CHCu[2%] layer as a function of time. One sees that the return shock reaches the inner surface of the sample layer at $t = 1.95$ ns, causing a density jump and shock heating to a temperature of $kT = 250$ eV; a heat wave follows the return shock due to strong thermal conduction from the hot core as a result of the large temperature gradient, leading to heating of the CHCu[2%] layer to $kT \approx 540$ eV (mid-panel of Fig. 5b); finally, the return shock approaches the outer surface of the sample layer at a later time of $t = 2.2$ ns.

Using these plasma conditions, we show the history of the Cu 2p-band population, as predicted by *VERITAS*, in the lower panel of Fig. 5b.

For a fully occupied *2p* band in Cu, there are six electrons in this band (energy level). The population of this *2p* band starts to deplete significantly at $t = 2.0$ ns when the heat wave raises the sample layer's temperature to over 300 eV, leading to the onset of *1s-2p* absorption, which is observed in the time-resolved spectra (Fig. 4e). Before this time, the fully occupied *2p* band does not allow *1s-2p* absorption to occur, so that $K_\alpha$ emission is the dominant feature in the x-ray spectra measured at early times (Fig. 4d).

The plasma conditions change throughout the sample layer as the return shock and heat wave propagate through the CHCu[2%] layer.

The observed spectrum represents this competition between $K_\alpha$ emission and shifted *1s-2p* absorption from different radial locations. Namely, the unshocked and colder regions give pronounced $K_\alpha$ emission, while the heated parts contribute dominantly to the *1s-2p* absorption feature. These processes compete in generating and transporting radiation and determine what is measured by the x-ray spectrometers.

Overall, the DFT-based *VERITAS* model reasonably describes the dynamic change in measured x-ray spectral features. The traditional *CRE* models might give the proper level of both $K_\alpha$ emission and *1s-2p* absorption, but their predictions tend to be highly dependent on their underlying atomic structure and continuum lowering models, which can make it difficult to isolate the physical effects of interest. For these high-adiabat and relatively-low velocity implosion studies on OMEGA, it is noted that the x-ray spectroscopy data are reproducible (see Supplementary Information).

To summarize, we have performed a theoretical and experimental study of atomic physics in Cu-doped plastic at several billion atmospheres of pressure. Overall, a DFT-based approach reproduces many of the emission and absorption features that are observed in the experiment, while traditional plasma spectroscopy treatments show sensitivity to the combination of atomic physics and continuum lowering models that are implemented. This sensitivity contributes to the present open questions on the validity of ad hoc continuum lowering models (see also ref. 54). This work indicates the necessity for a self-consistent treatment of dense plasma effects on altering atomic energy levels/bands and their populations at ultrahigh pressures. The DFT-based *VERITAS* approach, with potential future benchmarks using other buried metal and metal-alloy layers, could provide a reliable way for simulating radiation generation and transport in dense plasmas encountered in stars and inertial fusion targets. The experimental scheme reported here, based on a laser-driven implosion, can be readily extended to a wide range of materials in single- and multiple-shell geometries, opening the way for far-reaching investigations of extreme atomic physics and DFT models at tremendous pressures.

## Methods

### Implosion experiment on OMEGA
The experiments were conducted by symmetric laser drive using 60 Omega laser beams. Standard implosion diagnostics were used for these experiments, including neutron yield detector[55], wedge range filter for area-density measurement[56], and the neutron-timing diagnostic (NTD) for ion temperature and bang time[57].

### Measurement of x-ray emission spectra
Bragg reflection crystal spectrometers recorded time-integrated and time-resolved x-ray spectra in the energy range of 7800–8600 eV. One spectrometer[58] was coupled to an x-ray streak camera to achieve 80-ps time-resolution; the other was coupled to x-ray–sensitive image plate.

**Conversion to source emission.** From pinhole camera measurements of such implosions, the estimated x-ray source size is ~100-μm in diameter. With respect to the x-ray spectrometers which are 13–19.3 cm away from the target chamber center, the imploded capsule can be represented as a point source. The measured spectra were converted to source emission $S_\nu$ incident on each resolution element: $S_\nu \left[ \frac{ph}{sr \cdot eV} \right] = \frac{I_\nu}{f(E)T(E)G(E)}$, where $I_\nu$ is the measured signal density (photo-stimulated luminescence (PSL) per pixel for IP and analog-to-digital units (ADU) per pixel for the streak camera), $f(E)$ the instrument sensitivity function (signal (ADU or PSL) per photon), $T(E)$ the filter transmission, and $G(E) = R(E)\frac{dE}{d\theta}\frac{d\Omega}{dA}$ the spectrometer crystal response[59]. $f(E)$ is constructed from calibration measurements and detector models for both the IP[60] and streak camera[61,62]. The integrated

reflectivity $R(E)$ is calculated from the x-ray optics software XOP[63]. $\frac{dE}{d\theta}$ and $\frac{d\Omega}{dA}$ are calculated from a geometric ray-trace for each spectrometer.

Statistical uncertainty due to photon statistics in the time-integrated spectrometer is low, of order 0.5% after averaging over pixels in the non-dispersive dimension. In the time-resolved spectrometer, stochastic processes inherent to the streak camera amplification dominate statistical uncertainty, yielding ~30% fractional uncertainty after averaging over a resolution element of 80 ps. Systematic uncertainties include calibration measurements, filter thicknesses, and the crystal integrated reflectivity. The overall resolving power of the x-ray spectrometers is calibrated to be $E/\triangle E \approx 1100$. The resolving power of the x-ray spectrometers is primarily limited by source size broadening and crystal rocking curve effects[58].

**Streak camera time-base.** The time-base $t(x)$ was measured on separate shots by irradiating a Au foil with a train of laser pulses of known timing. The integration time of each pixel ("dwell time") $\Delta t = \frac{dt(x)}{dx}$ is used to calculate the source emission rate $\frac{dS_\nu}{dt}\left[\frac{ph}{sr \cdot eV \cdot s}\right] = \frac{S_\nu}{\Delta t}$. X-rays at the high energies of this spectrometer predominantly originate from the core; assuming the x-ray emission closely follows neutron production, the time-base is shifted to align the time of peak x-ray emission with the time of peak neutron production.

### Radiation-hydrodynamic simulations
The radiation-hydrodynamic simulations of implosion experiments have been performed with the 1-D *LILAC*[64] and 2-D *DRACO*[65] codes developed at the Laboratory for Laser Energetics. State-of-the-art physics models are employed in these simulations, including 3-D ray-tracing for laser energy deposition with a cross-beam energy transfer (CBET) model[66], the iSNB[67] nonlocal thermal transport model, and the first-principles equation-of-state (FPEOS[8,68,69]) and opacity tables (FPOT[70,71]) for constituent materials. For radiation energy transport, a multi-group diffusion model was used in *DRACO* with 48-group opacity tables. Cylindrical symmetry was enforced in the *2D DRACO* simulations, in which *r-z* coordinates are employed with the azimuthal symmetry axis along the *z*-axis. The *quasi-1D* nature of such high-adiabat implosions should lead to small 2D or 3D effects in x-ray emission calculations, justifying use of 2D (as opposed to 3D) simulations. Laser imprinting was simulated up to a maximum laser-speckle mode of $l = 150$ in *DRACO*, even though these simulations showed little effect of laser perturbations on such high-adiabat implosions with 1-ns square pulses at a laser intensity of $\sim 1.1 \times 10^{15}$ W cm$^{-2}$. *DRACO* simulation results were compared with experiments (e.g., see Table 2). The time-dependent 2-D density and temperature profiles, predicted from *DRACO* simulations, were used for further processing by a variety of *CRE* models and *VERITAS*, for x-ray spectral comparisons with experimental measurements.

### Collisional radiative equilibrium (*CRE*) modeling
Taking the *DRACO*-predicted plasma density and temperature profiles, we have applied the simulation package *Spect3D*[21] to perform the *CRE* modeling of x-ray spectra from these implosions. *Spect3D* uses atomic databases and continuum lowering models to track energy level populations, which are coupled to nonlocal radiation transport for x-ray generation (e.g., $K_\alpha$-emission of Cu), absorption, and propagation. The spectral resolving power ($E/\triangle E = 1100$), temporal resolution ($\delta t = 80$ ps), and spatial resolution ($\delta x = 10$ μm) were applied to the synthetic x-ray spectra from *Spect3D*. In *Spect3D* simulations, we processed 10 equal-angle-spaced radial lineouts of *DRACO*-predicted density and temperature profiles along different angular directions with kinetic models incorporating detailed atomic physics and radiation transport. We then averaged the resulting spectra among radial lineouts. Given the largely 1D-like performance of such high-adiabat

implosions (see Fig. 5a and Table 2), this quasi-2D treatment should be reasonable to avoid the time-consuming computations of 2D radiation transport with detailed atomic kinetics, for a relatively big 2D grid size of $601 \times 591$. Nevertheless, the use of 1-D radial lineouts in lieu of a full 2D or 3D analysis could be an additional cause for the small discrepancy observed in *VERTAS*-experiment comparisons.

## *VERITAS*: DFT-based multi-band kinetic modeling

The *VERITAS* code, developed in this work, is based on a density-functional theory (DFT) description of energy bands in dense plasma. The kinetic modeling of multi-band populations ($n_i$) is coupled with radiation transfer, as described by the following coupled equations for the steady state condition:

$$
\begin{cases}
\frac{dn_i}{dt} = -n_i \sum_{j \neq i}^{N} W_{ij}(I,\nu) + \sum_{j \neq i}^{N} n_j W_{ji}(I,\nu) = 0, \text{ for band } i \\
\mu \frac{\partial I(r,\nu)}{\partial r} + \frac{(1-\mu^2)}{r} \frac{\partial I(r,\nu)}{\partial \mu} = \eta(r,\nu) - \chi(r,\nu)I(r,\nu)
\end{cases}
\tag{1}
$$

with $W_{ij}$ being the transition rates among the total $N$ bands considered, which may depend on the specific intensity $I(r,\nu)$ of x-rays at radius $r$ and frequency $\nu$ (e.g., for photoionization and stimulated radiative processes). Here, the line of sight is along the $z$ axis, which has an angle $\theta$ relative to the 1-D spherical radial coordinate $r$, i.e., $\mu = \cos(\theta)$.

The above rate equation describes the population change for each band at the steady state condition ($dn/dt = 0$), due to radiative processes among the dipole-transition–allowed energy bands. For example, the population rate of change on band $i$ (due to radiative coupling to band $j$ with $E_i < E_j$) can be defined as the sum of the depopulating term $-n_i W_{ij} = -n_i B_{ij} \bar{I}_{ij}$ (stimulated absorption from $i$ to $j$) and the populating term $n_j W_{ji} = n_{ji} A_{ji} + n_{ji} B_{ji} \bar{I}_{ij}$ (spontaneous and stimulated emission from $j$ to $i$). Note that for the case of $E_i > E_j$ only the depopulating term appears in the rate equation for band $i$. Here, $n_{ij}$ and $n_{ji}$ are the maximum populations allowed for the corresponding radiative process. For instance, $n_{ji}$ will depend on the number of "*holes*" (depletion) in band $i$ and the weighted population in band $j$: $n_{ji} = \min[(n_{i,full} - n_i), (n_j \times g_i/g_j)]$ with $n_{i,full}$ being the fully-occupied population on band $i$ and $g_i(g_j)$ standing for the degeneracy of band $i$ ($j$). The Einstein coefficients $A$ and $B$ are related to the oscillator strength between bands $i$ and $j$, which can be calculated using DFT-determined orbitals. The frequency averaged mean radiation intensity is defined as $\bar{I}_{ij} = \int I(\nu) \times \phi_{ij}(\nu - \nu_{ij}) d\nu$, with the Voigt line profile $\phi_{ij}(\nu - \nu_{ij})$ centered at the frequency $\nu_{ij}$ corresponding to the energy gap between bands $i$ and $j$ and with line broadening models discussed later.

The emissivity $\eta(r,n,\nu)$ and absorption coefficient $\chi(r,n,\nu)$ have a dependence on the band population ($n$) of dense plasmas at the local grid $r$ and the radiation frequency $\nu$. The population changes on multiple energy bands are kinetically modeled by the rate equation (top equation), in which the radiative transition coefficients among different energy bands are calculated by using the DFT-determined orbitals, in contrast to the use of atomic databases of isolated atom plus continuum lowering in traditional *CRE* models. The DFT-based rate equation is then coupled with the radiation transfer equation (bottom equation) to simulate x-ray photoionization, emission, and band-band absorption processes throughout the density-temperature grid given by radiation-hydrodynamic codes. The radiation field at any spatial grid is determined by self-consistently solving the coupled rate and radiation transfer equations until a steady-state solution of the populations is achieved, similar to the procedure employed in *Spect3D*. Since the DFT description of energy bands in dense plasmas is self-consistent with the plasma environment, the band energy shift is naturally included for a given plasma condition. So, there is no need to invoke a continuum lowering model in *VERITAS*.

In principle, many energy bands can be included in *VERITAS*, without prescribing which are bound and which belong to the *continuum*, even though such a designation can be determined from density-of-state (DOS) calculations. Specifically, we have included the measurement-relevant energy bands [*1s, 2p, 3p*, and *continuum* of copper (Cu)] for modeling the implosion spectroscopy experiments. The main radiative processes considered among Cu's energy bands are: $1s \leftrightarrow continuum$ (photoionization/radiative-recombination), $1s \leftrightarrow 2p$ (band-band absorption/$K_\alpha$-emission), $1s \leftrightarrow 3p$ (band-band absorption/$K_\beta$-emission), $2p \leftrightarrow continuum$ (photoionization/radiative-recombination), and $3p \leftrightarrow continuum$ (photoionization/radiative-recombination). Even though we are focusing on the $1s \leftrightarrow 2p$ transition spectra, the inclusion of the bounded 3p-band of Cu is to ensure that all relevant population and depletion channels of the 1s-band are properly accounted for in the kinetic simulations. The exclusion of *2s* and *3s* bands from *VERITAS* modeling of the current experiments is based on the fact that their transitions to the *1s* band are dipole-forbidden and their coupling to *2p* and *3p* are outside the spectral range of interest. At the plasma conditions encountered here, the $n = 4$ bands of Cu have already merged into the *continuum*. For the conditions studied here ($20 - 500$ eV and $2 - 20\times$ solid-density), the rates for electron collisional processes are so high that local thermodynamic equilibrium is well maintained. Thus, one can take the thermal-DFT predicted band populations as a starting point in *VERITAS* to simulate the aforementioned radiative processes only, while the fast electron collisional processes are assumed to balance each other so that they can be omitted from current *VERITAS* simulations.

To enable the *VERITAS* simulations of these x-ray spectroscopy experiments, we have first built a DFT-based table storing the relevant frequency and oscillator strength of transitions among Cu's energy bands, for a density and temperature grid of CHCu[2%] spanning the mass density and temperature ranges of $\rho = 2 - 50$ g cm$^{-3}$ and $kT = 10 - 500$ eV. For each density and temperature condition, we have performed quantum molecular-dynamics (QMD) simulations to sample a variety of ionic configurations of dense CHCu[2%] plasma, based on the thermal-DFT formalism in either the Kohn-Sham orbital-based format or the orbital-free scheme. These DFT-based QMD calculations have been performed by using *ABINIT*[72] and our in-house OFMD code (*DRAGON*), with the Perdew-Burke-Ernzerhof (PBE) exchange-correlation functional[73]. For temperatures below ~50 eV the QMD simulations were done by using the orbital-based Kohn-Sham DFT implemented in *ABINIT*, while for high temperatures ($kT > 50$ eV) we turned to orbital-free DFT to perform our QMD simulations. Taking snapshots from QMD calculations, we performed the oscillator strength calculations for radiative transitions by using the Kubo-Greenwood package – KGEC@QUANTUM-ESPRESSO[74], with the *all-active-electron* projector-augmented-wave (PAW) potential[75]. These choices of DFT packages, exchange-correlation functional, and potentials follow standard practice in the warm-dense matter physics community. The band-energy deficiency from DFT calculations has been compensated with constant shifts derived from comparison with experimental energy levels of Cu at ambient condition. The band-energy "*deficiency*" from DFT refers to the small (~1–2%) difference between the DFT-calculated *1s-2p* energy gap of Cu and the experimentally measured Kα energy at ambient conditions. This small band-gap difference has been known as an *intrinsic* fact of DFT that uses approximated exchange-correlation functionals (e.g., PBE used here) which suffer from self-interaction error. These DFT calculations invoked 100–200 atoms of C, H, and Cu according to their atomic fractions in a supercell with periodic boundary conditions, which were converged with respect to number of bands, K-points, and energy cut-off.

With such a pre-calculated DFT table accessible to *VERITAS*, the radiative transition rates can be computed at any spatial grid for plasma conditions of CHCu[2%] given by rad-hydro simulations. It is

noted that the kinetic modeling was done only for the sample CHCu[2%] layer, while radiation transport in $D_2Ar$ and pure CH plasmas were calculated by using the emissivity and opacity tables from PrOpacEOS[76], same as what were used in *CRE* modeling. In principle, the band broadening of Cu can be determined from direct DFT-based QMD calculations. However, due to the limited number of Cu atoms involved in such demanding calculations, the resulting band-width (broadening) is currently not reliable due to the lack of sufficient sampling for charge-state distribution (CSD). Instead, we have adopted in *VERITAS* the temperature- and density-dependent broadening information coming from both SCRAM[22] and FAC[77] calculations for Stark (with an enhancement factor of ~5) and CSD broadening effects, as well as Doppler shift due to fluid motion, in a Voigt line profile. Both the SCRAM and FAC codes consider traditional plasma broadening mechanisms, including electron thermal-collision broadening[78], Stark broadening due to ion micro fields[79], and broadening from the charge-state distribution[80]. While all of these broadening mechanisms can explain the line-shape observations in low-density and high-temperature classical plasmas, they appear unable to account for the enhanced broadening seen in the dense plasmas created and reported here. We speculate that the current treatment of micro-field induced Stark broadening might have missed some of the density effects from coupled ions in such dense plasmas, hence the ad-hoc 5x increase in broadening applied to the *VERITAS* results. We hope the experimental observations of enhanced broadening reported here shall motivate future investigations on how density effects change line broadening in warm-dense plasmas. Finally, since *VERITAS* is based on the DFT description of dense plasmas, we expect its low-density limit to be around or slightly below ambient solid density, below which DFT calculations are no longer practically feasible and traditional atomic physics models should work better.

## Data availability

The experimental data, *Spect3D* simulation data, *VERITAS* simulation data that support the findings of this study are available from the corresponding authors upon request.

## Code availability

The *VERITAS* code that support the findings of this study are available from the corresponding authors upon request.

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

## Acknowledgements

This material is based upon work supported by the Department of
Energy National Nuclear Security Administration under Award Number
DE-NA0003856, the University of Rochester, and the New York State
Energy Research and Development Authority. This work is partially
supported by US National Science Foundation (NSF) PHY Grants No.
1802964 and No. 2205521 for SXH and VVK. The work of SBH was sup-
ported by Sandia National Laboratories' Laboratory Directed Research
and Development program under project 218456. Sandia National
Laboratories is a multimission laboratory managed and operated by
National Technology & Engineering Solutions of Sandia, LLC, a wholly
owned subsidiary of Honeywell International Inc., for the U.S. Depart-
ment of Energy's National Nuclear Security Administration under con-
tract DE-NA0003525. This paper describes objective technical results
and analysis. Any subjective views or opinions that might be expressed
in the paper do not necessarily represent the views of the U.S. Depart-
ment of Energy or the United States Government. Partial funding for
S.X.H., D.T.B. and P.M.N. was provided by the NSF Physics Frontier
Center Award PHY-2020249. This report was prepared as an account of
work sponsored by an agency of the U.S. Government. Neither the U.S.
Government nor any agency thereof, nor any of their employees, makes
any warranty, express or implied, or assumes any legal liability or
responsibility for the accuracy, completeness, or usefulness of any
information, apparatus, product, or process disclosed, or represents
that its use would not infringe privately owned rights. Reference herein
to any specific commercial product, process, or service by trade name,
trademark, manufacturer, or otherwise does not necessarily constitute
or imply its endorsement, recommendation, or favoring by the U.S.
Government or any agency thereof. The views and opinions of authors
expressed herein do not necessarily state or reflect those of the U.S.
Government or any agency thereof.

## Author contributions

S.X.H. and P.M.N. conceived the project and wrote the initial manuscript.
D.T.B., D.A.C., and P.M.N. performed the experimental measurements.
S.X.H. developed the DFT-based *VERITAS* code and performed the DFT
calculations, with help from I.E.G., V.V.K., S.B.H., D.I.M., N.R.S., and S.Z.
The *Spect3D* calculations were performed by M.G., I.E.G., and T.W.;
S.B.H. provided the SCRAM and *Muze* calculations. M.G. performed the
FAC calculations. All authors discussed the results and revised the
manuscript.

## Competing interests

The authors declare no competing interests.

## Additional information

**Supplementary information** The online version contains
supplementary material available at

S. X. Hu or Philip M. Nilson.

**Peer review information** *Nature Communications* thanks the anon-
ymous reviewer(s) for their contribution to the peer review of this
work. Peer reviewer reports are available.

