## [Peer Review File · Nature Communications]

Probing atomic physics at ultrahigh pressure using laser-driven implosionsREVIEWER COMMENTS

Reviewer #1 (Remarks to the Author):

Report on the article “Probing extreme atomic physics at Gbar pressure” by S.X. Hu et al.

Atomic physics in dense plasmas is an interesting subject of fundamental interest with some aspects located at the frontier of plasma physics and of condensed matter.

Standard calculations are based on the “chemical picture” in which atoms are seen as individual entities where energy levels, transitions rates, are calculated using standard atomic physics methods. These atomic data help to feed collisional-radiative models (CRM) to get populations kinetics of these energy levels. Plasma density effects are usually introduced by means of ad hoc Ionization Potential Depression (IPD) models. While used for a few decades, validity of this approach has been questioned recently in the context of solid density hot plasmas like those created by short intense x-ray pulse interaction with solid density matter.

This paper reports experiments and calculations concerning compressed matter(at pressures in the Gbar range) that go well beyond solid density and thus challenge the proper modeling of plasma spectral emission in this high-density regime. This is important for the spectroscopic interpretation of hot dense matter whether is belongs to astrophysical objects or laboratory plasmas. Experiments reported here are time-integrated and time-resolved spectroscopy data from laser-driven spherical implosions performed at the Omega laser facility. These data describe a complex situation where the outgoing spectrum evolves from $K\alpha$ emission to $1s-2p$ absorption. This evolution is directly related to the plasma conditions dynamically changing. Besides an interpretation based on the traditional approach (CRM+IPD), a new code (Veritas) based on condensed matter methods has been developed for a proper interpretation of these data. The capability of Veritas to model the outgoing experimental spectrum is impressive.

The paper is well-written and pleasant to read. The context is rather well presented and then describes a nice experiment to study x-ray fluorescence in compressed matter.

A clear conclusion is that we need to get rid of the standard collisional radiative models (which use ad hoc IPD models) for a spectroscopic analysis of this specific state of matter. The right alternative being a band-model adapted to condensed matter.

For all of these reasons, the article deserves to be published.

However, I have a few comments and remarks. I would like them to be adequately addressed before publication.

- I think that a short discussion on the modelling of near-edge absorption spectroscopy in warm dense matter, should be added. Indeed, some of these experiments revealing 1s-2p absorption features are also based on DFT calculations.

- Because of the large collection of models involved in Table 1 (not all of them being carefully described), I wonder if this table is really useful. Also, it rises the problem of the definition of Z^* . How is Z^* defined in the DFT models?

- My remaining remarks concern the description of the code Veritas developed in this work. I found the description of Veritas in the main text a bit overstated compared to what is discussed in the section Methods. I know details of this model cannot be given at length in the present paper and I encourage authors to publish them in another specific article. However, I think that present description of Veritas deserves some clarifications here. So, my questions concern mainly the part of the section Methods, dedicated to Veritas.

- A few details concerning the formulation of the transition rates W between 2 bands could be given.

- First equation of line 474, is not a right rate equation with populating and depopulating terms between bands, why?

- At the end what is the interest of this formulation since it said below that LTE prevails (although this is not so certain for $\rho = 2 \text{ g/cm}^2$ and $T_e = 500 \text{ eV}$).

- Justify the plane-parallel formulation of the Radiative Transfer equation.

- Articulation between Veritas and QMD (ABINIT) calculations as well as Orbital Free calculations deserves some more comments.

- Concerning the “all-active-electron projector-augmented-wave potential”, a proper reference should be given.

- What is the meaning of the “band-energy deficiency from DFT calculations”. I suppose this is related to the too high temperatures considered and to the number of bands that can practically be included in a calculation (and thus to the need of using an OFDFT model in some situations).

- More details concerning the broadening of a line between 2 bands should be given. In addition to the average over many atomic configurations (in space), do you consider some impact broadening by electrons. Justify the need to use an ad hoc broadening (with a factor 5 ?) from an independent model.

- Discuss the low-density limit of Veritas.

Reviewer #2 (Remarks to the Author):

Hu's article points to a possible solution to the continuum-lowering problem in atomic physics models using DFT with molecular dynamics. The article is very significant for the atomic physics of dense plasmas. However, several points should be answered or developed further before the article is published:

1. The results presented indicate that self-consistent models such as QMD are necessary to calculate the emission and absorption of radiation in warm plasmas. However, according to table I, the FAC code gives the closest results to the VERITAS code for temperatures of 200 and 300 eV, but FAC is not used in the comparisons in figure 1. In fact, the results in figure 3 are quite good for the FAC model with Ecker-Kroll, which the authors only point out as good for the case of XFEL-generated plasmas, as shown in reference 36. The rest of the comparisons are made with the ATBASE model, and its results are much worse than those of VERITAS. The authors should clarify why they have not used the FAC+EK model instead of ATBASE

2. Some readers might appreciate a few lines of explanation about the process of sphere compression by lasers, stagnation and emission during the flash-core.

3. Regarding the results of the 2D simulations (fig 5), it should be clarified which radiation energy transport model the code DRACO uses. It would also be useful to explain why the density and temperature variation is observed in figure 5b on the Z-axis and not on the X- or Y-axis (r-axis) and how it could affect the X-emission calculations.

Erratum:

line 194: implosion velocity should be 250 km/s

Reviewer #3 (Remarks to the Author):

Referee Comments regarding

"Probing extreme atomic physics at Gbar pressure" by S.X. Hu et al.

Overall: This manuscript describes novel, high-quality research which makes a

unique and significant contribution to understanding the behavior of dense plasmas in conditions specifically relevant to astrophysics (dwarf stars) and inertial confinement fusion, and more broadly relevant to quantum atomic/molecular physics generally. The paper includes both theoretical and experimental components and provides the first plausibly-sufficiently-realistic plasma-spectroscopy calculation at conditions which were thought, not so long ago, to be nearly intractable: the inner layers of an imploding shell at gigabar (millions of atmospheres) of pressure. This has potential to resolve a long-running debate about the competing "Stewart-Pyatt" and "Ecker-Kroll" models of continuum lowering (elimination of high-n atomic/ionic bound-electron states at high pressure).

The underlying work is certainly publishable, but this referee would like to see some significant revisions first - the manuscript as written could be substantially improved with careful attention in three broad areas:

- 1) The abstract and introduction should be significantly revised to be more specific and quantitative, to better "catch the reader's attention", and to make clear very quickly the particular (rather than general) contributions made in the work being reported. In many places the phrasing is overly vague and/or sentences do not follow clearly one-to-the next, making it hard for an outsider to understand the genuine significance of the work. Specific suggestions are provided in detail below.
- 2) As a Nature Communications paper, presumably the article is intended for a broader audience, not simply plasma spectroscopy specialists. With that in mind, key terms widely known within the field, but not more broadly, should be either defined in the text, or referenced more thoroughly. Some suggestions are provided below. Similarly, the logical flow of the paper should be explained up front and reinforced with either section boundaries or carefully phrased transition sentences between key paragraphs.

3) As written, the paper alludes to a series of experiments, but only one "laser shot" dataset is actually reported in any detail. This leaves open major questions about reproducibility, about potential trends (variation of key experimental parameters), and the possibility that the excellent fit to the data from the newly-developed VERITAS model might be a coincidence or a result of data over-fitting.

----- Specific Comments Below -----

Title: Gbar is not an SI unit. This could be off-putting to readers outside the authors' immediate field. Suggest rephrasing the title.

Within the text, please consider alternative units, or at least provide a handy SI conversion factor for readers who don't "speak gigabars" routinely.

Abstract - needs quantification; needs a stronger "hook" for general readers:

Line 18 - "Observable changes" isn't well explained. Change of what observable, due to what cause? For a general audience, "very high pressure" should be quantified.

Line 20 - Rephrase/Confusing: "dense astrophysical objects" are not observable or interpretable spectroscopically, since only the outer few units of optical depth is ever visible. Without "spectroscopic" the sentence still works.

Line 22 - "revealed inconsistencies" is too vague; it's not clear what is supposedly inconsistent with what.

Line 24-25: rephrasing of Line 22 is needed to motivate why general readers would care about "development of atomic and plasma physics models".

Line 26: rephrase: the *data* is not at several billion atmospheres; the plasma is.

Line 27: what's a "stagnating plastic shell?" (to a non-ICF-physics person)

One might imagine beads sitting on an algae-covered pond (of "stagnant" water)!

Expand and quantify the description of the shell and the experiment just a little.

Describe how it's possible in an implosion process for a material layer to both emit and absorb almost simultaneously. (This is explained deeper in the text, but just a little more detail here would really help readers get oriented.)

Line 28: rephrase: "K_α emission" and "1s-2p absorption" are related, near-inverse processes, which is very meaningful here, but this is obscured by the change in terminology. The description given of the K_α process later in the text would be more helpful here up front where the term is used for the first time.

Recommend using the 1s-2p formal language, rather than "K-alpha", since the latter is less broadly understood outside of spectroscopy community.

Line 28: Need to be more quantitative - HOW are the emission and absorption measured? Time, space, photon energy resolution? multiple sightlines? what parameters varied?

Line 29: replace "predictions" with "simulations". If shot data could potentially have informed the simulations (i.e., calculations are done after the shots) then the calculations are no longer predictions in the proper sense. "Pre-shot simulations" vs. "Post-shot simulations" would be clearer perhaps.

Line 31: If the development of self-consistent dense-plasma DFT is a major part of the work, that aspect should be explained in more detail. What does VERITAS add, or is it just a reassembly of existing methods and tools, put to a new purpose?

Line 32: Awkward phrasing. Could probably rewrite this sentence using half the words, and at most one comma, while retaining 100% of the meaning, by including or defining "CRE approach" within sentence on line 21, to avoid having to repeat "isolated atomic" and "ad hoc continuum lowering"... here.

Line 34-35: Weak finish. This is a vague throwaway hand-waving statement. "Dense plasma" and "Ultra-high pressure" have not been defined; "deeper understanding" lacks depth and clarity. Please quantify, be more specific, make connections to other work that will be impacted by this work... and/or say something that will make a Nature Communications reader want to know more.

Introduction:

The first sentence should say something more interesting.

The first paragraph should not only provide context but should also include the present work in that context. What impact will this paper be having?

Lines 61-97 appear to be a meandering rather than direct approach to the topic of the paper.

Somewhere in the introduction, authors should give reader some indication of the organization and logical flow of the paper.

Line 51: I don't believe there are any warm or dense plasmas at $T = 10^3$ K (0.1 eV)?

Are there actually dense *plasmas* in *planetary* cores? Examples? Reference?

Jupiter's core is typically described as a dense rock surrounded by liquid metallic hydrogen... which is not quite the same as a plasma.

Is ref 1, Burrows et al., a canonical reference for this material?

Planets and Brown dwarfs are not hot plasmas...

Line 53 "nascent stage" should either be justified or dropped

... given that white dwarf cosmochronometry has been around for 30-40 years...

Line 61 - rephrase: "in which" refers to plasmas, or to spectroscopy?

Line 65-66: rephrase (non-sequitur): What "has generally proven?"

(models aren't methods... models are plural and has is singular...)

Line 67: "CRE models" needs references; might also introduce the specific models used later in the paper.

Line 86: "non-ideal plasma" needs reference or definition

Line 88: "dilute", "cold" need quantification.

Line 93-95: need reference and/or quantification of conditions

Line 101: "DFT" needs definition and references.

Line 107: claim of "widely sought" needs support via references... or rephrase.

Line 109-111: This sentence is a main point of the introduction, and would be more impactful if it came significantly earlier and were rephrased to be more direct.

Line 112: Unexpected shift from third- to first-person. Suggest adopting consistent style.

Line 113-114 "multi-band kinetic model" needs elaboration.

Line 115: "thick shell" needs quantification

Line 119: Needs rephrasing. Experimental data cannot "reveal" similarities / differences between models. (Data can validate or invalidate models but similarities & differences between models are independent of any experiment.) However, comparison of model predictions for a given "experimentally inspired configuration" could reveal similarities/differences - without requiring any actual data from the experiment (an example of this is Figure 1).

Line 121-123: Rephrase - unsupported statements.

No "findings" have been provided yet in the text.

This sentence appears to be stating a conclusion without first providing any evidence. Suggest withholding claims of this type until later in the text, or else explaining where the supporting information will be found later in the text, or else quantifying and making more precise what "similarities and differences" were "revealed" in lines 119-120, that actually "demonstrate a need...".

Main text...

Line 124: Missing a transition... this seems to be an unexpected digression away from what referee thought was main thrust of the text... the prior paragraph is all about the experiment and its implications; why now switch to a theoretical a-priori toy model? furthermore the stated aim of this paragraph seems to be "to illustrate why" but the paragraph ends with no such demonstration.

Line 125: why is it called an "ice-block" model? Nothing in it is ice. Nor is it a block. If this is a specialized term, a reference would help. If ice-block is intended to be a new descriptive name, a bit more explanation about the choice of name would be helpful (for at least one reader...).

Line 139-140: need a less awkward transition. (perhaps "The remaining calculation in Fig 1b and 1c comes from VERITAS, a new DFT-based ...") The prior paragraph ends inconclusively otherwise, with discussion of Figure 1b and 1c including only the ATBASE CRE models.

Line 151 is redundant "our DFT-based kinetic code VERITAS"... all that already was said.

Line 154 - doesn't the exclusion of many electronic bands (line 146) from the model constitute a sort of "ad hoc continuum lowering" decision, by excluding consideration of higher-n states?

Lines 155-182 are well written and explain key spectral differences one might expect in but would be more digestible if broken up into a few paragraphs instead of a single page-long flow. Suggest breaking at line 164 "These differences..." and then again at Line 171 "To investigate what..."

Line 177 claims that 3 different DFT-based models are shown in Table 1. But only 2 are clearly listed in the table. The 2 CRE models are clearly stated.

Is FAC+AA considered a DFT-based model (on line 213 it's described as being CRE)?
Or are there only 2 DFT-based models here?

Also, since this is the first mention of FAC and SCRAM, and the AA model, some references and a bit of additional description of each of these approaches would be appropriate here.

Line 179 - what differences do the authors consider most significant and why?
(see also comments regarding Table I below)

Line 186 - more description of the Cu layer variations would be helpful.
How many options were tried? Which choices worked best? What happened to the other data?

Line 191 - what is "typical" here, given that a series of shots with varying Cu layer depths were apparently used? Which Cu location generated this data? How did the non-typical data vary from this one?

Line 205: More information is needed to support the gigabar pressure claim, and the warm-dense matter plasma conditions should be described in more detail than just pressure and density ranges. What temperature is predicted? How many shots were performed and what range of pressures were achieved (at least in simulation)?

This is a very interesting and novel experiment, which will pioneer a great deal of further work in this direction, but the claim that it's the first "precision spectroscopic measurement" from a gigabar-pressure high-density system is overly broad. For instance, implosions have been fielded for decades at comparable pressure, and many of those experiments used precision spectroscopy to make important

inferences (e.g. the MMI experiments on Ar-doped implosions, or others). Furthermore, the plasma conditions in the current experiment are only inferred from simulation, albeit one consistent with the new VERITAS calculations - but notably not consistent with the prior/current theoretical standard-bearers for inferring plasma conditions, which cannot fit the data well. A more skeptical referee might be concerned that the plasma conditions have more gradients or are otherwise not-as-simulated by DRACO etc, and that this might be why the standard tools break down. VERITAS could be an overfitting, or it could simply be best-matched to the EOS etc. used in the rad-hydro calculations to predict the time-dependent conditions.

Line 210 - the link to white dwarf stars is valuable, but also an example of how the abstract should be made more specific (it says "stellar interiors" and "dense astrophysical objects", which are overly broad compared to what is actually in the text.)

Line 212 - Following the comment about line 205 - it would help to have a better sense of the "DRACO-predicted dynamic plasma conditions", not merely at stagnation. What is the predicted density-temperature-pressure trajectory of the Cu-doped layer? What are the spatial gradients across the layer?

Line 232-233: what level of 3-D effects are possible and expected? Was the stalk in the line of sight of the diagnostics? Given the accuracy of the 2D simulations in predicting so much of the implosion performance, it appears contradictory to say that the simulation predicts the conditions accurately enough to model the spectrum from them, but then to claim that the conditions aren't as predicted by the simulation because of 3D effects. Some clarification would be helpful.

Lines 234-252: Given that many shots were performed, why is the analysis limited to just one of them? Can the others be summarized or used to provide support (perhaps via trends) to better refute a skeptical claim that the VERITAS model is somehow tuned to reproduce just the one shot? A single successful match is quite heroic for this type of plasma, but a really good model would match multiple shots, and conversely, one might learn quite a bit more from any trend discrepancies.

Line 286-287 - again, the "similarities and differences" should be described more precisely and quantitatively. Same density, different ionization...

Line 283-300: should say more, specifically, about VERITAS and its potential, as well as that of the experiments.

METHODS SECTION:

Line 420 claims a streak time resolution of 10 ps, but this is inconsistent with Line 461 which says simulations used 20 ps time resolution?

Line 422-423: please quantify (estimate based on available data) the observed X-ray source diameter.

Line 431-432 : quantify "low" and "substantially increased"

Line 434 - Is source broadening included in the calibration of the resolving power? What limits ΔE in the resolving power? Is there a published reference for the spectrometers being used?

Line 450 - typo - extra "." in $W \text{ cm}^{-2}$

Line 463-467: quantify how many radial lineouts were used and how they were chosen.

Line 467: This argument about using 1-D radial lineouts in lieu of a full 2- or 3-D analysis needs to be made more rigorous, and its potential inaccuracy in terms of key metrics should be quantified. For instance - Figure 5 shows a hot core of ~ 50 microns in diameter, and so the X-ray emission spot (including time-evolution not shown in Figure 5) must be at least as large. Meanwhile, depending on what is meant by the black arrows, the Cu layer may be just 50 microns away - one hot-spot diameter. Considering all raypaths which can reach a given spectrometer from across the hot-spot, there would seem to be some significant 2D effects possible - for instance, some rays will pass somewhat obliquely through the Cu layer, and therefore experience a different areal density. Perhaps this could also play into the question of experiment-VERITAS discrepancies which were discussed as possible "3D effects due to stalk" on lines 232-233?

Related: quantifying the typical grid spacing in the 2D draco simulations at stagnation time would be helpful.

Lines 470-534: Given that the VERITAS code is "developed in this work", it would be helpful to provide more information on the motivation & justification for specific choices in constructing the code. For example:

Why stop at $n=3$ if no ad-hoc assumptions are being made about continuum lowering?

Why exclude 2s and 3s states and 4+ states? While these may not be relevant to the specific transitions observed in the measurement, presumably they can still affect potential population distributions and thus rates? The calculated variation

in 2p populations suggests that 2s and 3s populations will also vary.

What motivated use of ABINIT vs. potential alternatives?

Ditto for PBE exchange-correlation functional - this choice would seem to be a critical one for accurate DFT.

Why is the Kubo-Greenwood package with all-active-electron PAW most appropriate?

Were any alternatives explored and if not, why not? If so, what issues were found that led to the current set of choices?

It would also increase credibility to show an array of validation/verification example calculations, and to illustrate more thoroughly the significant differences between this new approach and the other models.

Line 523-533: Please justify the apparently ad-hoc use of "5x enhanced" Stark broadening factors drawn from other codes? Why 5x? To a non-theoretical reader this smells like a tuning parameter which can be arbitrarily set to achieve a desired level of data-fitting. (A code whose name translates to "truth" might perhaps have an greater-than-usual justification to set a high bar for excellence!)

Line 490-491 "the band energy shift is naturally included" seems to be inconsistent with the line 529 comment that "the band-width is currently not reliable due to lack of sufficient sampling for charge-state distribution". Isn't it true that for $n=2$ transitions, both band positions and widths are critically dependent on the charge states that are present?

TABLES AND FIGURES:

Table 1 - appears to show errors - the electron densities presented cannot all be consistent with the fixed $\rho = 20 \text{ g/cc}$ and computed Z^* . (Unless ionization (Z^*) does not always create free electrons...) Is the paper asserting an error in some of the models?

Also: Z^* metric for a composite plasma (C, H, Cu) needs to be defined...

Figure 1 caption - typo "an Cu"

Figure 1 - annotation for the Cu-doped region is missing numerical density and temperature - this is inconsistent with the other annotations and needs to be explained more explicitly (e.g. "at 20 g/cm^3 and 200 or 300 eV"). Similarly caption for 1(c) could say "at mass density [...] and alternate temperature $kT = 300 \text{ eV}$ ".

Figure 2c: Need references. Consider including or at least mentioning relevant non-Hugoniot work by Doeppner et al on NIF (presented APS-DPP 2020).

Figure 2d: vertical axis needs more divisions. In current figure, reader cannot readily infer conditions probed by the current experiment without a ruler...

Also recommend stating the range of experimental conditions, with estimated uncertainties, in the figure caption. Is the range shown by the green oval intended to represent both uncertainty in the conditions as well as the range of variation within the experiment(s)? (If more than one experiment is included, need to change "experiment" to "experiments" in the caption.)

Also: The astrophysical notation for solar mass (e.g. "0.6M(.)" should be expanded to minimize potential for confusion among non-astrophysics readers.

Figure 3: Unless the calculations were actually done prior to the experiment (i.e. a blind-test comparison), "prediction" should be replaced with "calculations" or "results" etc.

To obtain time-integrated output from (static, equilibrium) CRE and DFT models, some range of time-dependent information must have been used. That should be explained.

For figures 3 and 4: The agreement between VERITAS and the experiment is sufficiently remarkable to warrant a separate graph overlaying the two curves and plotting the residual (difference between the two), which can be very informative regarding potential differences in the experimental vs. modeled physics...

Figure 5 (a) - needs redo - the color scale shown does not span full range of data presented (green is missing). In addition, red-green color scale is disadvantageous to roughly 1 in 15 male readers (colorblindness), suggest alternative colormap.

Also, the black arrow and notation on the graph are not explained and do not make sense. Both the inner and outer boundaries of the Cu doped layer at stagnation should be labeled. The stagnation time should be stated clearly in figure or caption and perhaps annotated in figure 5(b) as a dashed line for ease of comparison between the image and the time-evolution data.

5(b) - figure caption needs to explain which models' data were used to infer the Cu 2p populations (given the variation in Table 1...)

Response to Reviewers Comments on NCOMMS-22-15716

First of all, we would like to express our deep appreciation to the three expert reviewers who thoroughly assessed the work presented in the original manuscript. The suggestions and comments they provided are insightful and constructive for us to improve our manuscript for its clarity, conciseness, and attraction to a broad audience. We give below a point-by-point response to these comments along with our revisions [text in color “blue”] to the manuscript:

Response to Comments/Suggestions of Reviewer #1

We thank the reviewer for their positive remarks on the contents as “well-written and pleasant to read” and their excellent summary of the manuscript. The insightful comments/remarks given by the reviewer are extremely helpful and allow us to clarify the work and the results that are presented. For each of the remarks, we present our response and action below:

Point #1: *“I think that a short discussion on the modelling of near-edge absorption spectroscopy in warm dense matter, should be added. Indeed, some of these experiments revealing 1s-2p absorption features are also based on DFT calculations.”*

- **Response/Action:** Thank you for this good suggestion. We have added two sentences in the text to discuss the modeling of XANES in warm dense matter (WDM) based on DFT calculations, as well as three new references related to these discussions. It reads: “It is also noted that DFT-based modeling has been successfully applied to x-ray near-edge absorption spectroscopy (XANES) for warm-dense matter³⁶⁻³⁸. These earlier XANES experiments showed absorption features in good agreement with DFT calculations.³⁶⁻³⁸” The three new references are: [36]. Dorchies, F. & Recoules, V. Non-equilibrium solid-to-plasma transition dynamics using XANES diagnostic, *Phys. Rep.* **657**, 1 (2016); [37]. Jourdain, N., Lecherbourg, L., Recoules, V., Renaudin, P. & Dorchies, F. Ultrafast thermal melting in nonequilibrium warm dense copper, *Phys. Rev. Lett.* **126**, 065001 (2021), and [38]. Harmand, M. et al. X-ray absorption spectroscopy of iron at multimegabar pressures in laser shock experiments. *Phys. Rev. B* **93**, 024108 (2015).

Point #2: *“Because of the large collection of models involved in Table 1 (not all of them being carefully described), I wonder if this table is really useful. Also, it rises the problem of the definition of Z^* . How is Z^* defined in the DFT models?”*

- **Response:** The purpose of Table I is to help us put into perspective why we see what we see in Fig. 1 for a model system. Namely, what are the differences in physical quantities that drive the changes in spectral features between the two categories of models (DFT-based models vs. traditional collision-radiative models)? Yes, the question about the definition of Z^* in DFT is very tricky, even though one does NOT need to define it for DFT calculations. The reason to quote a Z^* for the DFT

calculations is that it is a widely used metric in the plasma physics community, even though there is no operator corresponding to Z^* in a quantum many-body theory like DFT. The Z^* quoted for the DFT models in Table I is calculated from the Thomas-Fermi average-atom model.

- **Action:** To clarify all of these, we have added the following sentences in the caption of Table I: “Comparisons of these predicted physical quantities demonstrate the differences between two categories of atomic physics models for warm- or hot-dense plasmas: DFT-based models vs. traditional collisional-radiative models. The quoted value of Z_{Cu}^* for VERITAS was calculated from the Thomas-Fermi average-atom model to provide a comparison to the other models, even though DFT calculations do not need to define Z_{Cu}^* . Additional details of these models can be found in the Methods section.”

Point #3: “My remaining remarks concern the description of the code Veritas developed in this work. I found the description of Veritas in the main text a bit overstated compared to what is discussed in the section Methods. I know details of this model cannot be given at length in the present paper and I encourage authors to publish them in another specific article. However, I think that present description of Veritas deserves some clarifications here. So, my questions concern mainly the part of the section Methods, dedicated to Veritas.”

- **Response:** We thank the reviewer for the encouragement! Yes, indeed we are planning to publish the details of VERITAS modeling in a technical article. Here, we’d like to clarify the technical details in the Methods section by following each of the reviewer’s remarks:

- **Remark/Question A:** “A few details concerning the formulation of the transition rates W between 2 bands could be given.”

- **Response/Action:** Yes, we take the reviewer’s suggestion and add the following sentences in the Methods section: “The above rate equation describes the population change for each band at the steady state condition ($dn/dt = 0$), due to radiative processes among the dipole-transition-allowed energy bands. For example, the population rate of change on band i (due to radiative coupling to band j with $E_i < E_j$) can be defined as the sum of the depopulating term $-n_i W_{ij} = -n_{ij} B_{ij} \bar{I}_{ij}$ (stimulated absorption from i to j) and the populating term $n_j W_{ji} = n_{ji} A_{ji} + n_{ji} B_{ji} \bar{I}_{ij}$ (spontaneous and stimulated emission from j to i). Note that for the case of $E_i > E_j$ only the depopulating term appears in the rate equation for band i . Here, n_{ij} and n_{ji} are the maximum populations allowed for the corresponding radiative process. For instance, n_{ji} will depend on the number of “holes” (depletion) in band i and the weighted population in band j : $n_{ji} = \min[(n_{i,full} - n_i), (n_j \times g_i/g_j)]$ with $n_{i,full}$ being the fully-occupied population on band i and g_i (g_j) standing for the degeneracy of band i (j). The Einstein coefficients A and B are related to the oscillator strength between bands i and j , which can be calculated using DFT-determined orbitals. The frequency averaged mean radiation intensity is defined as $\bar{I}_{ij} = \int I(\nu) \times \phi_{ij}(\nu -$

$v_{ij})dv$, with the Voigt line profile $\phi_{ij}(v - v_{ij})$ centered at the frequency v_{ij} corresponding to the energy gap between bands i and j and with line broadening models discussed later.”

- **Remark/Question B:** “*First equation of line 474, is not a right rate equation with populating and depopulating terms between bands, why?*”

- **Response/Action:** We feel the confusion might have been caused by the abbreviation notation used of the rate equation at the steady state condition ($dn/dt=0$). To clarify this, we have now re-written the equation in a more explicit way and clearly stated “**the steady state condition**”:

$$\frac{dn_i}{dt} = -n_i \sum_{j \neq i}^N W_{ij}(I, \nu) + \sum_{j \neq i}^N n_j W_{ji}(I, \nu) = 0, \text{ for band } i.$$

- **Remark/Question C:** “*At the end what is the interest of this formulation since it said below that LTE prevails (although this is not so certain for $\rho = 2 \text{ g/cm}^2$ and $T_e = 500 \text{ eV}$).*”

- **Response:** The reason we need the kinetic modeling is that Cu K-alpha emission relies on the photoionization of its $1s$ core electrons, even though the radiation field encountered by the hot-dense plasmas is still too weak to drive highly non-LTE populations. In the implosion experiments studied here, the Cu-doped CH layer never accesses low-density and high-temperature conditions like “ $\rho = 2 \text{ g/cm}^3$ and $T_e = 500 \text{ eV}$ ” quoted by the reviewer. Radiation-hydrodynamic simulations showed that the in-flight shell only reaches $\rho = 2\text{-}6 \text{ g/cm}^3$ and $T_e = 20\text{-}80 \text{ eV}$, while the plasma density jumps to $\rho = 10\text{-}20 \text{ g/cm}^3$ when the return shock and heat-wave reach the Cu-doped CH layer with temperatures rising to $200\text{-}500 \text{ eV}$. These dynamical conditions can be seen in Fig. 5b.

- **Remark/Question D:** “*Justify the plane-parallel formulation of the Radiative Transfer equation.*”

- **Response/Action:** We apologize for the confusion caused by the radiative transfer equation. In fact, our radiative transfer was done in spherical geometry with central symmetry, *i.e.*, depending on the radial coordinate r and $\mu = \cos(\theta)$. However, when we calculate the final radiation signal on the detector we have used the parallel line approximation (along z-direction) for projection. Now, in the revised manuscript the radiation transfer equation is correctly expressed in spherical geometry:

$$\mu \frac{\partial I(r, \nu)}{\partial r} + \frac{(1-\mu^2)}{r} \frac{\partial I(r, \nu)}{\partial \mu} = \eta(r, \nu) - \chi(r, \nu)I(r, \nu).$$

- **Remark/Question E:** “*Articulation between Veritas and QMD (ABINIT) calculations as well as Orbital Free calculations deserves some more comments.*”

- **Response/Action:** Yes, we have taken the reviewer’s suggestion and added the following sentences to the Methods section: “**For low temperatures below $\sim 50 \text{ eV}$ the QMD simulations were done by using the orbital-based Kohn-Sham DFT implemented**

in *ABINIT*, while for high temperatures ($kT > 50$ eV) we turned to orbital-free DFT to perform our QMD simulations.”

- **Remark/Question F:** “Concerning the “all-active-electron projector-augmented-wave potential”, a proper reference should be given.”
 - **Response/Action:** Yes, a new reference [77] is now added: Karasiev, V. V. and Hu, S. X. Unraveling the intrinsic atomic physics behind x-ray absorption line shifts in warm dense silicon plasmas. *Phys. Rev. E* **103**, 033202 (2021).

- **Remark/Question G:** “What is the meaning of the “band-energy deficiency from DFT calculations”. I suppose this is related to the too high temperatures considered and to the number of bands that can practically be included in a calculation (and thus to the need of using an OFDFT model in some situations).”
 - **Response/Action:** The band-energy “deficiency” from DFT refers to the small (~1-2%) difference between the DFT-calculated $1s-2p$ energy gap of Cu and the experimentally measured $K\alpha$ energy even at ambient condition. This small difference is a commonly known fact of DFT that uses approximated exchange-correlation functionals (e.g., PBE used here) with the suffering of *self-interaction* error. A constant energy shift, determined from the ambient-condition DFT calculation in comparison with experimental $K\alpha$ measurement of Cu, can be applied to other DFT calculations at different conditions. This procedure has proven to work well when compared to experiments. We found that such an *intrinsic* DFT deficiency (~1-2%) is slightly element-dependent, but does not vary with density and temperature for a specified material. To share such an observation with readers, we have added the following sentences in the Methods section: “The band-energy “deficiency” from DFT refers to the small (~1-2%) difference between the DFT-calculated $1s-2p$ energy gap of Cu and the experimentally measured $K\alpha$ energy at ambient conditions. This small band-gap difference has been known as an *intrinsic* fact of DFT that uses approximated exchange-correlation functionals (e.g., PBE used here) which suffer from *self-interaction* error.”

- **Remark/Question H:** “More details concerning the broadening of a line between 2 bands should be given. In addition to the average over many atomic configurations (in space), do you consider some impact broadening by electrons. Justify the need to use an ad hoc broadening (with a factor 5?) from an independent model.”
 - **Response/Action:** Yes, sampling the different atomic configurations has been included in the calculations. However, due to the limited number of Cu atoms used in such doped-target simulations, the DFT model does not produce the charge-state distribution as stated in the manuscript. The independent broadening model, coming from traditional plasma atomic physics calculations, is therefore needed to approximate the significant broadening due to charge state distribution, besides the usual Stark broadening by plasma electrons.

Nevertheless, we still found the broadening width given by this independent model underestimated the broadening width by a factor of ~ 5 , when compared to experiments for such dense plasmas. We have modified the discussion in the Methods section to reflect this: “It is noted that these traditional plasma atomic physics calculations have taken the broadening due to charge state distribution into account, in addition to the usual Stark broadening by plasma electrons. However, we still found that such calculations underestimate the total broadening width by a factor of ~ 5 , when compared to experiments for such dense plasmas.”

- **Remark/Question I:** “*Discuss the low-density limit of Veritas.*”
 - **Action:** A sentence has been added in the Methods section: “Finally, since VERITAS is based on the DFT description of dense plasmas, we expect its low-density limit to be around or slightly below ambient solid density, below which DFT calculations are no longer practically feasible and traditional atomic physics models should work better.”

With these detailed responses to Reviewer #1’s concerns and suitable changes made to the revised manuscript, we hope it can now be recommended for publication in Nature Communications.

Response to Comments/Suggestions of Reviewer #2

We thank the reviewer for remarking “*The article is very significant for the atomic physics of dense plasmas.*” The points raised are extremely helpful for us to clarify the manuscript. For each of these points, we give our response and action by following their order of appearance in the report:

Point #1: “*The results presented indicate that self-consistent models such as QMD are necessary to calculate the emission and absorption of radiation in warm plasmas. However, according to table I, the FAC code gives the closest results to the VERITAS code for temperatures of 200 and 300 eV, but FAC is not used in the comparisons in figure 1. In fact, the results in figure 3 are quite good for the FAC model with Ecker-Kroll, which the authors only point out as good for the case of XFEL-generated plasmas, as shown in reference 36. The rest of the comparisons are made with the ATBASE model, and its results are much worse than those of VERITAS. The authors should clarify why they have not used the FAC+EK model instead of ATBASE.*”

- **Response:** We thank the reviewer for pointing this out. We feel the confusion was probably caused by the lack of detailed description of the “FAC+AA” model listed in Table I, for calculating Z_{Cu}^* and $2p$ -populations. This “FAC+AA” model involves the atomic structure calculation of a Cu atom that is “embedded” into the CH plasma mixture in which the plasma environment (free-electron density) is self-consistently described by the average-atom (AA) model. It mostly resembles the DFT+QMD treatment of such a plasma mixture; that’s why it gives the closest results to DFT+QMD in Table I, for the specific plasma conditions listed

there. However, this new development of “FAC+AA” (using an average-atom model to describe the plasma environment) has not yet been implemented into the integrated Spect3D modeling software. Instead, the available versions of Spect3D only have the capability to apply an FAC-calculated database with traditional EK or SP continuum lowering models. This is why in Fig.3 we can only compare with “FAC+EK” and “FAC+SP” simulations, although we expect “FAC+AA” would probably give similar results as VERITAS (and experiments). Back to Fig.3, it is likely that the similarity of “FAC+EK” to experiment and VERITAS is coincidental (even though it still overestimated the K-alpha peak by ~50%), as the HED-physics community is now aware that the traditional continuum lowering models of Ecker-Kroll has limited physical basis. Namely, the EK model (or SP) cannot be generalized to all dense plasma systems. To give an example, an experimental group, who originally advocated EK for explaining ionization-potential-depression (IPD) of XFEL-heated Al-plasma [Nature 482, 59 (2012)], has recently published new IPD measurements of other solid-density pure and compound plasmas in *Nat. Commun.* 7, 1173 (2016) (DOI: 10.1038/ncomms11713). These new measurements indicate little change in IPD between different compounds, unequivocally contradicting predictions of the EK model as shown below (taken from the paper):

Figure 4: IPD inferred by the K-edge measurements. The reduction in the ionization potential of Mg, Al and Si in the different materials is plotted as a function of the ionic charge state and is compared with the predictions of analytical models (EK and SP—see text), assuming a plasma ionization equal to the charge state. This figure is taken from the following reference: Ciricosta, O. et al. *Nat. Commun.* 7, 11713 (2016).

- **Action:** To clarify the confusion of “FAC+AA” and “FAC+EK”, we have added the following sentences in the text: “The “FAC+AA” model uses FAC-code calculations for the atomic structure of a Cu atom that is “embedded” in a CH plasma mixture in which the plasma environment is described by an average-atom (AA)-type model. It embodies a similar “*spirit*” to DFT with a self-consistent-field (SCF) calculation of plasma screening for an atom embedded in a plasma mixture.” “The resemblance between the FAC+Ecker-Kroll model (Fig. 3d) and experiments is likely coincidental, as other recent measurements⁵³ of ionization-potential-depression have defied the Ecker-Kroll model.”

Point #2: “Some readers might appreciate a few lines of explanation about the process of sphere compression by lasers, stagnation and emission during the flash-core.”

- **Response/Action:** We have taken the reviewer’s suggestion and added the following sentences in the text: “When the laser pulse irradiates the spherical capsule, laser ablation launches a strong shock wave that compresses the target. After the shock breaks out of the inner surface of the shell into the gas-filled core, the shell is accelerated inwards until it stagnates at a certain radius. At stagnation, the contained gas is compressed and heated to form a hot core, which emits x-rays that probe the stagnating shell and enable our spectroscopic measurements.”

Point #3: “Regarding the results of the 2D simulations (fig 5), it should be clarified which radiation energy transport model the code DRACO uses. It would also be useful to explain why the density and temperature variation is observed in figure 5b on the Z-axis and not on the X- or Y-axis (r-axis) and how it could affect the X-emission calculations.”

- **Response/Action:** Again, we take the reviewer’s points and have made the following descriptions of DRACO simulation details in the Methods section/Radiation-hydrodynamics simulations: “For radiation energy transport, a multi-group diffusion model was used in DRACO with 48-group opacity tables. Cylindrical symmetry was enforced in the 2D DRACO simulations, in which *r-z* coordinates are employed with the azimuthal symmetry axis along the *z*-axis. The *quasi-1D* nature of such high-adiabat implosions should lead to small 2D or 3D effects in x-ray emission calculations, justifying use of 2D (as opposed to 3D) simulations.”

Erratum: “line 194: implosion velocity should be 250 km/s”.

- **Response/Action:** Thank you for catching this typo; It has been corrected in the revised manuscript: the unit of “cm s⁻¹” has been changed to “km s⁻¹”.

With these detailed responses to Reviewer #2’s points and corresponding clarifications made, we hope the revised manuscript can now be recommended for publication in Nature Communications.

Response to Comments/Suggestions of Reviewer #3

We are deeply appreciative and feel encouraged by the reviewer's positive remarks on our manuscript as "...*novel, high-quality research which makes a unique and significant contribution to understanding the behavior of dense plasmas...*". With their professional insights and dedication to help, the Reviewer identified three specific areas for us to clarify. Without any doubt, we believe that addressing these concerns and suggestions has significantly improved the clarity and conciseness of our manuscript. For that we cannot thank the Reviewer enough!

Point #1: "*The abstract and introduction should be significantly revised to be more specific and quantitative, to better "catch the reader's attention", and to make clear very quickly the particular (rather than general) contributions made in the work being reported. In many places the phrasing is overly vague and/or sentences do not follow clearly one-to-the next, making it hard for an outsider to understand the genuine significance of the work. Specific suggestions are provided in detail below.*"

- **Response/Action:** We agree with the reviewer and have rewritten the abstract and introduction to better "*catch the reader's attention.*" Detailed modifications are listed below based on the reviewer's comments.

Point #2: "*As a Nature Communications paper, presumably the article is intended for a broader audience, not simply plasma spectroscopy specialists. With that in mind, key terms widely known within the field, but not more broadly, should be either defined in the text, or referenced more thoroughly. Some suggestions are provided below. Similarly, the logical flow of the paper should be explained up front and reinforced with either section boundaries or carefully phrased transition sentences between key paragraphs.*"

- **Response/Action:** We thank the reviewer for this comment. Taking these suggestions, we have thoroughly revised the manuscript by following these specific instructions provided by the reviewer. Details of modifications made to the revised text are given below.

Point #3: "*As written, the paper alludes to a series of experiments, but only one "laser shot" dataset is actually reported in any detail. This leaves open major questions about reproducibility, about potential trends (variation of key experimental parameters), and the possibility that the excellent fit to the data from the newly-developed VERITAS model might be a coincidence or a result of data over-fitting.*"

- **Response:** This is another important and insightful point raised by the reviewer. We take the responsibility for having not discussed the experimental reproducibility in the original manuscript. Indeed, these experimental campaigns have lasted for three years on OMEGA; many repeatable shots have been conducted. All of such high-adiabat and relatively-lower implosion velocity experiments are robust and repeatable. The *VERITAS* simulations for these

shots also reproduce the major spectroscopic features observed in experiments. For illustration purpose, we show the time-integrated spectral measurements in the following, for another shot similar to what presented in the original manuscript:

- **Action:** To clarify this point and to share the discussions with readers, we have added a paragraph in the new section of “Discussion”. It reads as “For these high-adiabat and relatively-low velocity implosion studies on OMEGA, it is noted that the x-ray spectroscopy data are highly reproducible (see Supplementary Information).”

-----Response/Action to Specific Comments Below-----

Again, we thank the reviewer for their detailed comments on our manuscript. We find all of them constructive and helpful to improve the paper. To better present our responses and actions, we group related comments together below:

Reviewer’s Specific Comment #1: “Title: Gbar is not an SI unit. This could be off-putting to readers outside the authors' immediate field. Suggest rephrasing the title. Within the text, please consider alternative units, or at least provide a handy SI conversion factor for readers who don't "speak gigabars" routinely.”

- ❖ **Response/Action:** Yes, we have rephrased the title to be “Probing atomic physics at ultra-high pressures”. We have now provided an SI conversion factor for gigabar: (1 gigabar = 10^{14} Pa), as suggested by the reviewer.

Reviewer’s Specific Comment #2: “Abstract - needs quantification; needs a stronger "hook" for general readers: Line 18 - "Observable changes" isn't well explained. Change of what observable,

due to what cause? For a general audience, "very high pressure" should be quantified; Line 20 - Rephrase/Confusing: "dense astrophysical objects" are not observable or interpretable spectroscopically, since only the outer few units of optical depth is ever visible. Without "spectroscopic" the sentence still works; Line 22 - "revealed inconsistencies" is too vague; it's not clear what is supposedly inconsistent with what. Line 24-25: rephrasing of Line 22 is needed to motivate why general readers would care about "development of atomic and plasma physics models". Line 26: rephrase: the *data* is not at several billion atmospheres; the plasma is. Line 27: what's a "stagnating plastic shell?" (to a non-ICF-physics person) One might imagine beads sitting on an algae-covered pond (of "stagnant" water)!"

- ❖ **Response/Action:** We took the reviewer's suggestion and rewrote the starting sentences in the abstract. Now, they read as: **"Spectroscopic measurements of dense plasmas at billions of atmospheres (i.e., billions of times the pressure at the Earth's surface) provide tests of our fundamental understanding of how matter behaves at extreme conditions – and by extension the interpretation of dense astrophysical objects such as white dwarf stars. Developing reliable atomic physics models at these conditions, benchmarked by experimental data, is crucial to an improved understanding of radiation transport in both stars and inertial fusion targets. However, detailed spectroscopic measurements at these conditions are rare, and traditional collisional-radiative equilibrium (CRE) models, based on isolated-atom calculations and *ad hoc* continuum lowering models, have proved questionable at and beyond solid density – leaving open the possibility for more-accurate methods. Here we report time-integrated and time-resolved x-ray spectroscopy measurements at several billion atmospheres (Gbar) using a laser-driven spherical implosion."**

Reviewer's Specific Comment #3: "Expand and quantify the description of the shell and the experiment just a little. Describe how it's possible in an implosion process for a material layer to both emit and absorb almost simultaneously. (This is explained deeper in the text, but just a little more detail here would really help readers get oriented.) Line 28: rephrase: "K_alpha emission" and "1s-2p absorption" are related, near-inverse processes, which is very meaningful here, but this is obscured by the change in terminology. The description given of the K_alpha process later in the text would be more helpful here up front where the term is used for the first time. Recommend using the 1s-2p formal language, rather than "K-alpha", since the latter is less broadly understood outside of spectroscopy community. Line 28: Need to be more quantitative - HOW are the emission and absorption measured? Time, space, photon energy resolution? multiple sightlines? what parameters varied?"

- ❖ **Response/Action:** As suggested by the reviewer, the sentence in Line 28 has been rewritten: **"We use the imploding shell and its hot core (kT ~ keV) at stagnation to probe a Cu-doped witness layer located inside the shell, inducing Cu K α emission and 1s-2p absorption."**

Reviewer's Specific Comment #4: "Line 29: replace "predictions" with "simulations". If shot data could potentially have informed the simulations (i.e., calculations are done after the shots) then the calculations are no longer predictions in the proper sense. "Pre-shot simulations" vs. "Post-shot simulations" would be clearer perhaps."

- ❖ **Response/Action:** We agree and have changed “**radiation-hydrodynamic predictions**” to “**post-shot radiation-hydrodynamic simulations.**”

Reviewer’s Specific Comment #5: *“Line 31: If the development of self-consistent dense-plasma DFT is a major part of the work, that aspect should be explained in more detail. What does VERITAS add, or is it just a reassembly of existing methods and tools, put to a new purpose? Line 32: Awkward phrasing. Could probably rewrite this sentence using half the words, and at most one comma, while retaining 100% of the meaning, by including or defining “CRE approach” within sentence on line 21, to avoid having to repeat “isolated atomic” and “ad hoc continuum lowering”... here.”*

- ❖ **Response/Action:** Yes, we agree and have rewritten the sentence in Line 32: “**These measurements, augmented by experimentally constrained post-shot radiation-hydrodynamic simulations of the imploded plasma conditions, are in good agreement with a self-consistent treatment of the dense-plasma environment based on density-functional theory (DFT), which is developed in this work.**”

Reviewer’s Specific Comment #6: *“Line 34-35: Weak finish. This is a vague throwaway hand-waving statement. “Dense plasma” and “Ultra-high pressure” have not been defined; “deeper understanding” lacks depth and clarity. Please quantify, be more specific, make connections to other work that will be impacted by this work... and/or say something that will make a Nature Communications reader want to know more.”*

- ❖ **Response/Action:** We have rewritten the concluding sentence in the abstract. It now reads: “**These results indicate the necessity and viability of modeling dense plasmas with self-consistent methods like DFT, which impact the accuracy of radiation transport simulations used to describe stellar evolution and the design of inertial fusion targets.**”

Reviewer’s Specific Comment #7: *“Introduction: The first sentence should say something more interesting. The first paragraph should not only provide context but should also include the present work in that context. What impact will this paper be having?”*

- ❖ **Response/Action:** We have rewritten the first sentence: “**The physics of warm and hot dense matter can unravel the mysterious inner workings of planetary cores and stellar interiors.**” To place the present work in context, we have added the following sentence at the end of the first paragraph: “**The implosion spectroscopy measurements and model development presented in this work aim to reveal a more-detailed picture of atomic physics in dense-plasma environments at billion atmosphere (Gbar) pressures.**”

Reviewer’s Specific Comment #7: *“Lines 61-97 appear to be a meandering rather than direct approach to the topic of the paper. Somewhere in the introduction, authors should give reader some indication of the organization and logical flow of the paper.”*

- ❖ **Response/Action:** The second paragraph (lines 61-79) sets the stage by describing the status of spectroscopic measurements in HED physics. We have added section/subsection

titles as well as the following sentences at the end of the introduction to provide a logical flow to the paper: “The paper is organized as follows: first, the necessity of a reliable atomic physics model for interpreting x-ray spectroscopic measurements is demonstrated using a surrogate dense-plasma object. The experimental results are then presented with a detailed spectral comparison between measurements and simulations based on traditional atomic physics models and the DFT-based approach that is developed in this work. Finally, the implications of these results for understanding dense plasma environments are discussed.”

Reviewer’s Specific Comment #8: “Line 51: I don't believe there are any warm or dense plasmas at $T = 10^3$ K (0.1 eV)? Are there actually dense *plasmas* in *planetary* cores? Examples? Reference? Jupiter's core is typically described as a dense rock surrounded by liquid metallic hydrogen... which is not quite the same as a plasma. Is ref 1, Burrows et al., a canonical reference for this material? Planets and Brown dwarfs are not hot plasmas...”

- ❖ **Response:** In the high-energy-density (HED) physics community, the use of “warm-dense matter” and “warm-dense plasmas” is getting indistinguishable these days. For planetary cores such as the outer core of Earth and Jupiter’s interior (whether or not there is a solid core inside Jupiter is still debatable), the temperatures can get to above ~0.5 eV to a few eV so that free electrons can appear in such warm-dense situation.

Reviewer’s Specific Comment #9: “Line 53 “nascent stage” should either be justified or dropped ... given that white dwarf cosmochronometry has been around for 30-40 years...Line 61 - rephrase: “in which” refers to plasmas, or to spectroscopy? Line 65-66: rephrase (non-sequitur): What “has generally proven?” (models aren't methods... models are plural and has is singular...). Line 67: “CRE models” needs references; might also introduce the specific models used later in the paper.”

- ❖ **Action:** All these suggestions are taken. The phrase “nascent stage” has been removed. The sentences referred to are rewritten: “X-ray spectroscopy, a common and sometimes only means to diagnose and understand short-lived plasmas, measures x-ray emission and absorption with spatial, spectral, and/or temporal resolution¹²⁻¹⁶.” “Reliable atomic and plasma physics models are required to interpret these spectral signatures and have generally proven to be adequate for spectroscopically diagnosing classical/ideal plasmas¹⁷⁻²⁰.” CRE model references have been added: “collision-radiative equilibrium (CRE) models^{21,22}”.

Reviewer’s Specific Comment #10: “Line 86: “non-ideal plasma” needs reference or definition; Line 88: “dilute”, “cold” need quantification. Line 93-95: need reference and/or quantification of conditions. Line 101: “DFT” needs definition and references. Line 107: claim of “widely sought” needs support via references... or rephrase. Line 101: “DFT” needs definition and references. Line 107: claim of “widely sought” needs support via references... or rephrase.”

- ❖ **Action:** All of these terms are now defined or quantified following the reviewer’s suggestions: “non-ideal (i.e., strongly-coupled and/or degenerate)”; “the dilute, but cold ($n_e = 10^{15} - 10^{18}$ cm⁻³ and $T = 10^3 - 10^5$ K)”; “density-functional theory (DFT)”; “are widely sought because of” has been rephrased to “very important because of”.

Reviewer's Specific Comment #11: *“Line 109-111: This sentence is a main point of the introduction, and would be more impactful if it came significantly earlier and were rephrased to be more direct.”*

- ❖ **Response/Action:** We thank the reviewer for this suggestion. This sentence has been rephrased and moved to the first paragraph in the introduction. It reads: “Spherically-convergent techniques uniquely access the gigabar pressure regime in experiments, providing the necessary data to test atomic physics models for warm and hot dense plasmas.”

Reviewer's Specific Comment #12: *“Line 112: Unexpected shift from third- to first-person. Suggest adopting consistent style. Line 113-114 "multi-band kinetic model" needs elaboration. Line 115: "thick shell" needs quantification.”*

- ❖ **Action:** All three points are taken: “Here, we report x-ray spectroscopy measurements at gigabar pressures using laser-driven implosions.” The following sentence has been added to elaborate the multi-band kinetic model: “The VERITAS model uses DFT-derived band (atomic level) information to compute the radiative transition rates that can be coupled to the radiation transfer equation to describe the radiation generation and transport processes in a dense plasma.” “thick shell” is now quantified: “30- μ m-thick plastic shell.”

Reviewer's Specific Comment #13: *“Line 119: Needs rephrasing. Experimental data cannot "reveal" similarities / differences between models. (Data can validate or invalidate models but similarities & differences between models are independent of any experiment.) However, comparison of model predictions for a given "experimentally inspired configuration" could reveal similarities/differences - without requiring any actual data from the experiment (an example of this is Figure 1).”*

- ❖ **Action:** This sentence is now rephrased: “..., allowing the spectroscopic data to differentiate the DFT-based kinetic model from traditional treatments based on isolated-atom calculations and *ad hoc* continuum-lowering models.”

Reviewer's Specific Comment #14: *“Line 121-123: Rephrase - unsupported statements. No "findings" have been provided yet in the text. This sentence appears to be stating a conclusion without first providing any evidence. Suggest withholding claims of this type until later in the text, or else explaining where the supporting information will be found later in the text, or else quantifying and making more precise what "similarities and differences" were "revealed" in lines 119-120, that actually "demonstrate a need...”*

- ❖ **Response/Action:** We agree with the reviewer on this point. The following sentence has been removed, as this concluding statement is repeated in the Discussion section: “These findings demonstrate the need for a self-consistent description of warm dense plasma environments, providing direct insight into the microphysical response of matter at much higher pressures than were previously possible.”

Reviewer's Specific Comment #15: *"Line 124: Missing a transition... this seems to be an unexpected digression away from what referee thought was main thrust of the text... the prior paragraph is all about the experiment and its implications; why now switch to a theoretical a-priori toy model? furthermore the stated aim of this paragraph seems to be "to illustrate why" but the paragraph ends with no such demonstration."*

- ❖ **Action:** We added the following section and subsection titles to make the transition: **“Results
Surrogate dense-plasma object.”** This style of section/subsection transition is recommended by Nature Communications.

Reviewer's Specific Comment #16: *"Line 125: why is it called an "ice-block" model? Nothing in it is ice. Nor is it a block. If this is a specialized term, a reference would help. If ice-block is intended to be a new descriptive name, a bit more explanation about the choice of name would be helpful (for at least one reader...)."'*

- ❖ **Response/Action:** The specialized term of “ice-block” has been widely used in the ICF and HED plasma physics community. It refers to a “toy” plasma object that consists of *blocks* of constant density and temperature plasmas, for the purpose of testing plasma physics models. To avoid *irritation* (for at least one reader), we have changed the phrase “ice-block model” to “**surrogate dense-plasma object.**”

Reviewer's Specific Comment #17: *"Line 139-140: need a less awkward transition. (perhaps "The remaining calculation in Fig 1b and 1c comes from VERITAS, a new DFT-based ...") The prior paragraph ends inconclusively otherwise, with discussion of Figure 1b and 1c including only the ATBASE CRE models."*

- ❖ **Response/Action:** Again, we thank the reviewer for this excellent suggestion. We have now expanded the referee-suggested sentence for the transition at the end of that paragraph: **“The remaining results in Figs. 1b and 1c come from VERITAS, a new DFT-based multi-band kinetic model for dense plasma spectroscopy.”**

Reviewer's Specific Comment #18: *"Line 151 is redundant "our DFT-based kinetic code VERITAS"... all that already was said."*

- ❖ **Response:** The repeating statement here aims to emphasize the fundamental difference between traditional CRE models and this DFT-based approach. We may of course remove it if the reviewer strongly dislikes it.

Reviewer's Specific Comment #19: *"Line 154 - doesn't the exclusion of many electronic bands (line 146) from the model constitute a sort of "ad hoc continuum lowering" decision, by excluding consideration of higher-n states?"'*

- ❖ **Response:** The choice of how many bands to be included in the kinetic modeling does NOT constitute any sort of “continuum lowering” decision. This is because once the self-consistent-field DFT calculation determines all of the bands (eigen-energy levels of the

many-body system under mean-field approximation), we do NOT need to define which bands are “discrete” and which bands are “continuum” (even though the density-of-state [DOS] can tell you those bands having structureless DOS belong to the “continuum”)! In principle, one can include all these bands into *VERITAS* for kinetic modeling of radiative transition processes. However, for the 1s-2p spectral range probed by the current experiment we can focus on the major radiative-transition “channels” (and their involved bands) to simplify the kinetic modeling without losing important physics.

Reviewer’s Specific Comment #20: *“Lines 155-182 are well written and explain key spectral differences one might expect in but would be more digestible if broken up into a few paragraphs instead of a single page-long flow. Suggest breaking at line 164 “These differences...” and then again at Line 171 “To investigate what...”*

- ❖ **Action:** Taking the reviewer’s suggestion, we have broken this long paragraph into three in the revised manuscript.

Reviewer’s Specific Comment #21: *“Line 177 claims that 3 different DFT-based models are shown in Table 1. But only 2 are clearly listed in the table. The 2 CRE models are clearly stated. Is FAC+AA considered a DFT-based model (on line 213 it’s described as being CRE)? Or are there only 2 DFT-based models here? Also, since this is the first mention of FAC and SCRAM, and the AA model, some references and a bit of additional description of each of these approaches would be appropriate here.”*

- ❖ **Response:** We apologize for the confusion between “FAC+AA” and “FAC+Stewart-Pyatt/Ecker-Kroll”. We feel the confusion was probably caused by the lack of a detailed description of the “FAC+AA” model listed in Table I, for calculating Z^* and $2p$ -populations. This “FAC+AA” model involves the atomic structure calculation for a Cu atom that is “embedded” into the CH plasma mixture in which the plasma environment (free-electron density) is self-consistently described by the average-atom (AA) model. It mostly resembles the DFT+QMD treatment of such a plasma mixture; that’s why it gives the closest results to DFT+QMD in Table I, for the specific plasma conditions listed there. Therefore, we have considered “FAC+AA” as one of DFT-based model. However, this new development of “FAC+AA” has not yet been implemented into the integrated modeling software -- Spect3D. Instead, the available versions of Spect3D have only capabilities of applying FAC-calculated atomic database with traditional Stewart-Pyatt (SP) or Ecker-Kroll (EK) continuum lowering models. This is why FAC+SP and FAC+EK used in Fig. 3 are still traditional CRE models.
- ❖ **Action:** To clarify the confusion of “FAC+AA” and “FAC+SP/EK”, we have added the following sentences in the text: “The “FAC+AA” model uses FAC-code calculations for the atomic structure of a Cu atom that is “embedded” in a CH plasma mixture in which the plasma environment is described by an average-atom (AA)-type model. It embodies a similar “spirit” to DFT with a self-consistent-field (SCF) calculation of plasma screening for an atom embedded in a plasma mixture.”

Reviewer's Specific Comment #22: *“Line 179 - what differences do the authors consider most significant and why? (see also comments regarding Table I below)”*

- ❖ **Response:** The $2p$ population is the most important difference, since it determines whether the $1s-2p$ transition is allowed (e.g., absorption is forbidden if $2p$ is fully occupied) as well as the transition rate amplitude.

Reviewer's Specific Comment #23: *“Line 186 - more description of the Cu layer variations would be helpful. How many options were tried? Which choices worked best? What happened to the other data? what is "typical" here, given that a series of shots with varying Cu layer depths were apparently used? Which Cu location generated this data? How did the non-typical data vary from this one?”*

- ❖ **Response:** Information on experimental reproducibility have been added into the new Discussion section and Supplementary Information (see the above response to Reviewer's point #3). We found the typical 3- μm Cu-layer depth gave the best data, meaning the appearance of both emission and absorption features.

Reviewer's Specific Comment #24: *“Line 205: More information is needed to support the gigabar pressure claim, and the warm-dense matter plasma conditions should be described in more detail than just pressure and density ranges. What temperature is predicted? How many shots were performed and what range of pressures were achieved (at least in simulation)? Line 212 - Following the comment about line 205 - it would help to have a better sense of the "DRACO-predicted dynamic plasma conditions", not merely at stagnation. What is the predicted density-temperature-pressure trajectory of the Cu-doped layer? What are the spatial gradients across the layer?”*

- ❖ **Action:** We have added the following sentence to give experimental plasma temperature and pressure conditions inferred from simulations: “Inferred from DRACO simulations, the plasma temperature and density conditions in the imploding Cu-doped layer vary from \$kT \approx 10-50\$ eV and \$\rho \approx 2-10\$ g \$\text{cm}^{-3}\$ (in-flight stage) to \$kT \approx 200-500\$ eV and \$\rho \approx 10-25\$ g \$\text{cm}^{-3}\$ during stagnation. The corresponding pressure in the compressed shell changes from \$\sim 50\$ Mbar to a maximum value approaching \$\sim 5\$ Gbar.”

Reviewer's Specific Comment #25: *“This is a very interesting and novel experiment, which will pioneer a great deal of further work in this direction, but the claim that it's the first "precision spectroscopic measurement" from a gigabar-pressure high-density system is overly broad. For instance, implosions have been fielded for decades at comparable pressure, and many of those experiments used precision spectroscopy to make important inferences (e.g. the MMI experiments on Ar-doped implosions, or others). Furthermore, the plasma conditions in the current experiment are only inferred from simulation, albeit one consistent with the new VERITAS calculations - but notably not consistent with the prior/current theoretical standard-bearers for inferring plasma conditions, which cannot fit the data well. A more skeptical referee might be concerned that the plasma conditions have more gradients or are otherwise not-as-simulated by DRACO etc, and that*

this might be why the standard tools break down. VERITAS could be an overfitting, or it could simply be best-matched to the EOS etc. used in the rad-hydro calculations to predict the time-dependent conditions.”

- ❖ **Action:** This statement has now been modified to be more specific: “our experiment has extended the pressure and density conditions at which both time-integrated and time-resolved x-ray spectroscopic measurements have been conducted: gigabar (Gbar) pressures and $\sim 15 - 20 \times$ solid-density, as indicated by Fig. 2c.”

Reviewer’s Specific Comment #26: “Line 210 - the link to white dwarf stars is valuable, but also an example of how the abstract should be made more specific (it says "stellar interiors" and "dense astrophysical objects", which are overly broad compared to what is actually in the text.)”

Action: The words like "stellar interiors" and "dense astrophysical objects" in the abstract have been modified to be specific link to “white dwarf stars”.

Reviewer’s Specific Comment #27: “Line 232-233: what level of 3-D effects are possible and expected? Was the stalk in the line of sight of the diagnostics? Given the accuracy of the 2D simulations in predicting so much of the implosion performance, it appears contradictory to say that the simulation predicts the conditions accurately enough to model the spectrum from them, but then to claim that the conditions aren't as predicted by the simulation because of 3D effects. Some clarification would be helpful.”

- ❖ **Response:** Over the past two decades, it is widely recognized by the ICF physics community that high-adiabat ($\alpha = 6-10$) implosions with thick shells are less susceptible to laser and target perturbations than high-speed and low-adiabat ($\alpha = 2-3$) fusion implosions. High-adiabat implosion performance can be largely captured by 1D and 2D radiation-hydrodynamic simulations. This is the reason why thick targets were chosen for the present spectroscopic study. 3D effects (no matter how small they are) cannot be completely ruled out, even if they do not significantly affect the time-integrated measurements of neutron yield, ion temperature, and areal density. These time-integrated measurements could still “miss” some temporal and spatial variations in the imploding shell, while could be revealed by more sensitive x-ray spectroscopic measurements. One such 3D perturbation is the localized stalk, which might potentially “push” a small portion of Cu-doped layer closer to the hot spot. To some extent, this small part of “hotter” Cu-doped layer will contribute to the overall x-ray spectra different from the majority of the target, especially for the case in which the stalk is unavoidably seen by the broad view of our x-ray spectrometers.
- ❖ **Action:** To clarify this confusion, we have re-written this sentence: “This small spectroscopic discrepancy might be attributed to some unavoidable three-dimensional effects, even though the time-integrated implosion measurements overall agree with 2D DRACO simulations. For instance, the stalk perturbation could inject a small and localized portion of the Cu-doped layer closer to the hot spot, which, to some extent, could contribute to the measured spectra in ways that are not accounted for in the 2D model.”

Reviewer's Specific Comment #28: *“Lines 234-252: Given that many shots were performed, why is the analysis limited to just one of them? Can the others be summarized or used to provide support (perhaps via trends) to better refute a skeptical claim that the VERITAS model is somehow tuned to reproduce just the one shot? A single successful match is quite heroic for this type of plasma, but a really good model would match multiple shots, and conversely, one might learn quite a bit more from any trend discrepancies.”*

- ❖ **Response:** This point has been addressed in the response to reviewer's point #3 above.

Reviewer's Specific Comment #29: *“Line 286-287 - again, the "similarities and differences" should be described more precisely and quantitatively. Same density, different ionization...”*

- ❖ **Action:** This sentence has been removed.

Reviewer's Specific Comment #30: *“Line 283-300: should say more, specifically, about VERITAS and its potential, as well as that of the experiments.”*

- ❖ **Response/Action:** Yes, we took the reviewer's suggestion and added the following sentence in the Discussion: *“The DFT-based VERITAS approach, with potential future benchmarks using other buried metal and metal-alloy layers, could provide a reliable way for simulating radiation generation and transport in dense plasmas encountered in stars and inertial fusion targets.”*

METHODS SECTION:

Reviewer's Specific Comment #31: *“Line 420 claims a streak time resolution of 10 ps, but this is inconsistent with Line 461 which says simulations used 20 ps time resolution?”*

- ❖ **Response:** This typo has been fixed; the time-resolution is ~80-ps from careful check of spectrometer calibration. Figure 4 has also been updated with 80-ps resolution.

Reviewer's Specific Comment #32: *“Line 422-423: please quantify (estimate based on available data) the observed X-ray source diameter.”*

- ❖ **Action:** This info has been added: *“From pinhole camera measurements of such implosions, the estimated x-ray source size is ~100- μ m in diameter. With respect to the x-ray spectrometers which are 13-19.3 cm away from the target chamber center, the imploded capsule can be represented as a point source.”*

Reviewer's Specific Comment #33: *“Line 431-432 : quantify "low" and "substantially increased”*

- ❖ **Action:** These quantifications are done; the new sentence is now reading as: *“Statistical uncertainty due to photon statistics in the time integrated spectrometer is low, of order 0.5% after averaging over pixels in the non-dispersive dimension. In the time-resolved*

spectrometer, stochastic processes inherent to the streak camera amplification dominate statistical uncertainty, yielding ~30% fractional uncertainty after averaging over a resolution element of 80ps.”

Reviewer’s Specific Comment #34: *“Line 434 - Is source broadening included in the calibration of the resolving power? What limits delta-E in the resolving power? Is there a published reference for the spectrometers being used?”*

- ❖ **Response/Action:** Yes, source broadening has been included in the resolving power calibration. The following sentence and reference have been added: “The resolving power of our x-ray spectrometers is primarily limited by source size broadening and crystal rocking curve⁵⁸.” The new reference is: Millecchia, M. et al. Streaked x-ray spectrometer having a discrete selection of Bragg geometries for Omega. *Rev. Sci. Instrum.* **83**, 10E107 (2012).

Reviewer’s Specific Comment #35: *“Line 450 - typo - extra ”. in $W\text{ cm}^{-2}$. Line 463-467: quantify how many radial lineouts were used and how they were chosen.”*

- ❖ **Response/Action:** The typo has been fixed. We added the lineout info: “10 equal-angle-spaced radial lineouts”.

Reviewer’s Specific Comment #36: *“Line 467: This argument about using 1-D radial lineouts in lieu of a full 2- or 3-D analysis needs to be made more rigorous, and it's potential inaccuracy in terms of key metrics should be quantified. For instance - Figure 5 shows a hot core of ~50 microns in diameter, and so the X-ray emission spot (including time-evolution not shown in Figure 5) must be at least as large. Meanwhile, depending on what is meant by the black arrows, the Cu layer may be just 50 microns away - one hot-spot diameter. Considering all raypaths which can reach a given spectrometer from across the hot-spot, there would seem to be some significant 2D effects possible - for instance, some rays will pass somewhat obliquely through the Cu layer, and therefore experience a different areal density. Perhaps this could also play into the question of experiment-VERITAS discrepancies which were discussed as possible "3D effects due to stalk" on lines 232-233? Related: quantifying the typical grid spacing in the 2D draco simulations at stagnation time would be helpful.”*

- ❖ **Response/Action:** We fully agree with the reviewer that the use of 1-D radial lineouts in lieu of a full 2- or 3-D analysis could be an additional cause for the small discrepancy we saw in VERITAS-experiment comparison. We have now tried our best to justify this choice and pointed out the potential inaccuracy this might have caused, by rewriting the sentences: “Given the largely 1D-like performance of such high-adiabat implosions (see Fig. 5a and Table 2), this quasi-2D treatment should be reasonable to avoid the time-consuming computations of 2D radiation transport with detailed atomic kinetics, for a relatively big 2D grid size of 601×591 . Nevertheless, the use of 1-D radial lineouts in lieu of a full 2D or 3D analysis could be an additional cause for the small discrepancy observed in VERITAS-experiment comparisons.”

Reviewer's Specific Comment #37: *"Lines 470-534: Given that the VERITAS code is "developed in this work", it would be helpful to provide more information on the motivation & justification for specific choices in constructing the code. For example: Why stop at $n=3$ if no ad-hoc assumptions are being made about continuum lowering? Why exclude 2s and 3s states and 4+ states? While these may not be relevant to the specific transitions observed in the measurement, presumably they can still affect potential population distributions and thus rates? The calculated variation in 2p populations suggests that 2s and 3s populations will also vary."*

- ❖ **Response/Action:** We have taken the reviewer's suggestion and added the following sentences in the Methods section to give more information: "In principle, many energy bands can be included in VERITAS, without prescribing which are bound and which belong to the continuum, even though such a designation can be determined from density-of-state (DOS) calculations." "The exclusion of 2s and 3s bands from VERITAS modeling of the current experiments is based on the fact that their transitions to the 1s band are dipole-forbidden and their coupling to 2p and 3p are outside the spectral range of interest. At the plasma conditions encountered here, the $n=4$ bands of Cu have already merged into the continuum."

Reviewer's Specific Comment #38: *"What motivated use of ABINIT vs. potential alternatives? Ditto for PBE exchange-correlation functional - this choice would seem to be a critical one for accurate DFT. Why is the Kubo-Greenwood package with all-active-electron PAW most appropriate? Were any alternatives explored and if not, why not? If so, what issues were found that led to the current set of choices? It would also increase credibility to show an array of validation/verification example calculations, and to illustrate more thoroughly the significant differences between this new approach and the other models."*

- ❖ **Response/Action:** These DFT packages and choices of exchange-correlation functional and potentials have been widely accepted by the warm-dense matter physics community, as they provide the most efficient and accurate-enough calculations to generally agree with experiments. To share this info with readers, we have added the following sentence: "These choices of DFT packages, exchange-correlation functional, and potentials follow standard practice in the warm-dense matter physics community."

Reviewer's Specific Comment #39: *"Line 523-533: Please justify the apparently ad-hoc use of "5x enhanced" Stark broadening factors drawn from other codes? Why 5x? To a non-theoretical reader this smells like a tuning parameter which can be arbitrarily set to achieve a desired level of data-fitting. (A code whose name translates to "truth" might perhaps have an greater-than-usual justification to set a high bar for excellence!)"*

- **Response/Action:** If we have infinite computing power, DFT calculations with large number of atoms in the computation box should be able to sample the line broadening in the system. However, due to the limit number of Cu atoms used in such doped-target simulations that we can afford currently, we have missed the charge-state distribution as stated in the manuscript. The independent broadening model, coming from traditional plasma atomic physics calculations, is therefore needed for taking that major effect into account, besides the usual Stark broadening by plasma electrons. Nevertheless, we still

found the broadening width given by this independent model was still underestimated the broadening width by a factor of ~ 5 , when compared to experiments for such dense plasmas. We have modified the discussion in the section of Methods to reflect this: “It is noted that these traditional plasma atomic physics calculations have taken the broadening due to charge state distribution into account, besides the usual Stark broadening by plasma electrons. However, we found that such calculations still underestimated the total broadening width by a factor of ~ 5 , when compared to experiments for such dense plasmas.”

Reviewer’s Specific Comment #40: “Line 490-491 “the band energy shift is naturally included” seems to be inconsistent with the line 529 comment that “the band-width is currently not reliable due to lack of sufficient sampling for charge-state distribution”. Isn’t it true that for $n=2$ transitions, both band positions and widths are critically dependent on the charge states that are present?”

- ❖ **Response:** The band energy shift is determined by the averaged charge state in the system that DFT calculation can give, while the accurate charge state distribution requires large number of atoms in the simulations box. The latter is missing in our current simulations the limit number of Cu atoms used in such doped-target simulations that we can afford currently.

TABLES AND FIGURES:

Reviewer’s Specific Comment #41: “Table 1 - appears to show errors - the electron densities presented cannot all be consistent with the fixed $\rho = 20$ g/cc and computed Z^* . (Unless ionization (Z^*) does not always create free electrons...) Is the paper asserting an error in some of the models? Also: Z^* metric for a composite plasma (C, H, Cu) needs to be defined...”

- ❖ **Response/Action:** Yes, the total electron density should be same for a certain mass density. However, what we compared is “free-electron density” which depends on the averaged ionization degree of the whole plasma of CHCu[2%]. We feel sorry about the confusion of Z^* : here we meant to compare Z^* of Cu in the CH plasma mixture, not Z^* for the whole CHCu[2%] plasma. To clarify this, we have now changed “ Z^* ” to “ Z_{Cu}^* ” in Table I.

Reviewer’s Specific Comment #42: “Figure 1 caption - typo “an Cu”; Figure 1 - annotation for the Cu-doped region is missing numerical density and temperature - this is inconsistent with the other annotations and needs to be explained more explicitly (e.g. “at 20 g/cm³ and 200 or 300 eV”). Similarly caption for 1(c) could say “at mass density [...] and alternate temperature $kT = 300$ eV.”

- ❖ **Action:** All of these have now been fixed by following the reviewer’s suggestion.

Reviewer’s Specific Comment #43: “Figure 2c: Need references. Consider including or at least mentioning relevant non-Hugoniot work by Doepfner et al on NIF (presented APS-DPP 2020). Figure 2d: vertical axis needs more divisions. In current figure, reader cannot readily infer conditions probed by the current experiment without a ruler...Also recommend stating the range of experimental conditions, with estimated uncertainties, in the figure caption. Is the range shown

by the green oval intended to represent both uncertainty in the conditions as well as the range of variation within the experiment(s)? (If more than one experiment is included, need to change "experiment" to "experiments" in the caption.) Also: The astrophysical notation for solar mass (e.g. "0.6M(.)" should be expanded to minimize potential for confusion among non-astronomy readers."

- ❖ **Action:** All of these suggestions are taken. According modifications are done.

Reviewer's Specific Comment #44: "Figure 3: Unless the calculations were actually done prior to the experiment (i.e. a blind-test comparison), "prediction" should be replaced with "calculations" or "results" etc. To obtain time-integrated output from (static, equilibrium) CRE and DFT models, some range of time-dependent information must have been used. That should be explained."

- ❖ **Response/Action:** Yes, the word "prediction" has been replaced by "calculations". The following sentence has been added into the caption of Fig. 3: "The time integration in calculations has been done from $t=1.7$ ns to $t=2.4$ ns during the hot-spot flash, with snapshots for each 20-ps time interval."

Reviewer's Specific Comment #45: "For figures 3 and 4: The agreement between VERITAS and the experiment is sufficiently remarkable to warrant a separate graph overlaying the two curves and plotting the residual (difference between the two), which can be very informative regarding potential differences in the experimental vs. modeled physics..."

- ❖ **Response/Action:** We have added this suggested residual plot for the time-integrated spectra and its related discussions into Supplementary Information. For the time-resolved ones, Figs. 4d, 4e, and 4f have aligned VERITAS and experiments together in each individual figure. Thus, we think there is no need to plot the residual between them.

Reviewer's Specific Comment #46: "Figure 5 (a) - needs redo - the color scale shown does not span full range of data presented (green is missing). In addition, red-green color scale is disadvantageous to roughly 1 in 15 male readers (colorblindness), suggest alternative colormap. Also, the black arrow and notation on the graph are not explained and do not make sense. Both the inner and outer boundaries of the Cu doped layer at stagnation should be labeled. The stagnation time should be stated clearly in figure or caption and perhaps annotated in figure 5(b) as a dashed line for ease of comparison between the image and the time-evolution data. 5(b) - figure caption needs to explain which models' data were used to infer the Cu 2p populations (given the variation in Table 1...)"

- ❖ **Response/Action:** We thank the reviewer for many of these good "catches"! We have redone the figure and captions by following what the reviewer suggested.

With these detailed responses to Reviewer #3's points and suitable changes made to the revised manuscript, we hope it can now be recommended for publication in Nature Communications.

REVIEWER COMMENTS

Reviewer #1 (Remarks to the Author):

Dear authors and Editors,

I thank the authors for the answers to my comments and for the modifications.

There remains just one point whose response is not fully satisfactory. This point concerns the broadening problem and the ad hoc factor 5 which is needed. First, one comment, SCRAM and FAC are just codes including (apparently) an electron broadening model. To be fair, this (these) electron broadening model(s) deserve(s) direct citation. Now, what about ion Stark broadening in these codes? In principle, due to ionization and thermal agitation, there exist in plasmas, a microscopic electric field (often considered as quasi-static) that splits energy levels. Dedicated plasma broadening codes consider this effect. Could this effect be the source of the ad hoc factor 5 or explain partially this factor? It would be nice that the authors dig a bit this question or give more justifications for this factor 5.

Best regards.

Reviewer #2 (Remarks to the Author):

The article has now become much clearer and more precise. The answers to the referees are adequate and consistent. I think it is suitable for publication.

Reviewer #3 (Remarks to the Author):

Referee appreciates the considerable effort the authors have made to improve the manuscript for a general audience, while also responding to the detailed physics questions by the 3 referees.

There are a handful of relatively minor points which were not fully addressed and may (editors' discretion) be worth a little further effort:

1) Suggest moving or copying "2p->1s transition" from Line 165 to Line 114 (where K_a emission is first mentioned), to clarify for non-spectroscopic experts that K_a emission is the inverse process of 1s->2p absorption. It's significant / novel / interesting that both of these inverse processes are observed on the same experiment, but from different causes and this could be called out more clearly.

2) Table 2 header should say "experiment" (no s) since only one shot is included. Alternatively, including the data from the replication shot 97615 in this table appears to be possible given the available space (presumably the DRACO simulation wouldn't need to be redone for a near-identical repeat shot), and would strengthen the results.

3) Line 197 Experimental Setup and/or Figure 2a: Need to specify the placement of the 10um thick Cu-doped layer within the 30um thick shell. Figure 2a indicates that there is pure CH both inside and outside the Cu layer, but not how much.

4) Line 321 and/or Supplementary Information: The second shot data is highly appreciated but the claim that "data are highly reproducible" is somewhat overstated. 2 shots from the same day are better than one, but not sufficient to make claim that "data are highly reproducible". Reproducibility over multiple shot days and N>2 shots might be expected from a "3 year experiment". How many other shots were done that show similar levels of agreement with the 2 example shots presented in the text? Authors comment in Supplementary Note 1 that that "multiple" experiments were done over a few years, but without quantification it's not clear if "multiple" means 2, 20, or 200 shots. There's no other data, experimental parameter variations or other pertinent information given in the text. It would also not be difficult to quantify the "high degree of repeatability" claimed in the text: range of variation or standard deviations of neutron yields for successful shots; of integrated X-ray signals, etc. It would increase the value of the VERITAS match to the data to provide just a few more bits of information to support the claims of experimental reproducibility.

Response to Reviewers Comments on NCOMMS-22-15716A

We thank all three reviewers for comments on our revised manuscript. The suggestions are insightful and constructive for us to further improve our manuscript for its clarity. We give below a point-by-point response to these comments along with our revisions [text in color “blue”] to the manuscript:

Response to Comments of Reviewer #1

We thank the reviewer for being satisfied by most of our last responses to his/her comments and the corresponding modifications made to the revised manuscript. There is only one “*not-fully-satisfactory*” concern left for us to further clarify:

Reviewer #1’s Point: *“This point concerns the broadening problem and the ad hoc factor 5 which is needed. First, one comment, SCRAM and FAC are just codes including (apparently) an electron broadening model. To be fair, this (these) electron broadening model(s) deserve(s) direct citation. Now, what about ion Stark broadening in these codes? In principle, due to ionization and thermal agitation, there exist in plasmas, a microscopic electric field (often considered as quasi-static) that splits energy levels. Dedicated plasma broadening codes consider this effect. Could this effect be the source of the ad hoc factor 5 or explain partially this factor? It would be nice that the authors dig a bit this question or give more justifications for this factor 5.”*

- **Response:** We thank the reviewer for their comments. To be honest, we believe electron energy level broadening in dense plasma has not been well understood. The experimental data provide compelling evidence for future development of a reliable broadening theory for dense plasmas. Yes, the SCRAM and FAC codes consider all of the traditional plasma broadening mechanisms, including electron thermal-collision broadening, Stark broadening due to micro-fields from other ions, as well as broadening from the charge-state distribution. Taken the reviewer’s suggestion, direct citations to these broadening mechanisms have now been added as new references [80-82]. While these broadening mechanisms can explain the line-shape observations in low-density and high-temperature classical plasmas, they seem unable to account for the enhanced broadening seen in dense plasmas created by the experiments reported here. We suspect that the current treatment of micro-field—induced Stark broadening might have missed some of the density effects from coupled ions in such dense plasmas. We hope the experimental observations reported here will motivate future investigations on how density effects change line broadening in warm-dense plasmas.

Action: To share these frank discussions with readers, we have modified the paragraph on page 23. It now reads: “Both the SCRAM and FAC codes consider traditional plasma broadening mechanisms, including electron thermal-collision broadening⁸⁰, Stark broadening due to ion micro fields⁸¹, and broadening from the charge-state distribution⁸². While all of these broadening mechanisms can explain the line-shape observations in low-density and high-temperature classical plasmas, they appear unable to account for the enhanced broadening seen in the dense plasmas created and reported here. We speculate that the current treatment of micro-field induced Stark broadening might have missed some of the density effects from coupled ions in such dense plasmas, hence the *ad-hoc* 5x increase in broadening applied to the VERITAS results. We hope the experimental observations of enhanced broadening reported here shall motivate future investigations on how density effects change line broadening in warm-dense plasmas.”

With the response and suitable addition made, we hope the revised manuscript can now be accepted for publication in *Nature Communications*.

Response to Comment of Reviewer #2

We thank the reviewer for recommending our revised manuscript for publication.

Response to Comments/Suggestions of Reviewer #3

We are grateful to see that Reviewer #3 was generally satisfied by our last responses and modifications made to the revised manuscript. We are more than happy to further address the specific four points raised by the Reviewer:

Reviewer #3's Point 1: “Suggest moving or copying “2p->1s transition” from Line 165 to Line 114 (where K_{α} emission is first mentioned), to clarify for non-spectroscopic experts that K_{α} emission is the inverse process of 1s->2p absorption. It's significant / novel / interesting that both of these inverse processes are observed on the same experiment, but from different causes and this could be called out more clearly.”

- **Response/Action:** Yes, this suggestion is taken and we have copied “2p->1s transition” to Line 114. We have also added the following sentence on page 6: “Both of these inverse processes are observed on the same experiment; photo-ionization of 1s electrons enables K_{α} -emission, and thermal-ionization of 2p electrons enables 1s-2p absorption.”

Reviewer #3's Point 2: “Table 2 header should say "experiment" (no s) since only one shot is included. Alternatively, including the data from the replication shot 97615 in this table appears to be possible given the available space (presumably the DRACO simulation wouldn't need to be redone for a near-identical repeat shot), and would strengthen the results.”

- **Response/Action:** Suggestion is taken: “experiments” and “simulations” have been changed to “experiment” and “simulation” in the header of Table 2.

Reviewer #3's Point 3: “Line 197 Experimental Setup and/or Figure 2a: Need to specify the placement of the 10um thick Cu-doped layer within the 30um thick shell. Figure 2a indicates that there is pure CH both inside and outside the Cu layer, but not how much.”

- **Response/Action:** We thank the reviewer for catching the omission of the detailed target description. To give this information, we have added the following sentence on page 10: “The 10- \$\mu\$ m-thick Cu-doped layer begins \$\sim\$ 3- \$\mu\$ m from the inner surface of the CH shell.”

Reviewer #3's Point 4: “Line 321 and/or Supplementary Information: The second shot data is highly appreciated but the claim that "data are highly reproducible" is somewhat overstated. 2 shots from the same day are better than one, but not sufficient to make claim that "data are highly reproducible". Reproducibility over multiple shot days and $N > 2$ shots might be expected from a "3 year experiment". How many other shots were done that show similar levels of agreement with the 2 example shots presented in the text? Authors comment in Supplementary Note 1 that that "multiple" experiments were done over a few years, but without quantification it's not clear if "multiple" means 2, 20, or 200 shots. There's no other data, experimental parameter variations or other pertinent information given in the text. It would also not be difficult to quantify the "high degree of repeatability" claimed in the text: range of variation or standard deviations of neutron yields for successful shots; of integrated X-ray signals, etc. It would increase the value of the VERITAS match to the data to provide just a few more bits of information to support the claims of experimental reproducibility.”

- **Response/Action:** We agree with the reviewer on this assessment about missing information of experimental variations over the three-year campaign on OMEGA. We now explicitly provide this information by adding the following paragraph to the Supplementary Information: “For these spherical implosion spectroscopy campaigns on OMEGA, we have scanned some key experimental parameters, including laser pulse shape (1ns and 2-ns square pulse), the distance of the Cu-dopant layer from the inner surface of the shell (3 to 10 \$\mu\$ m), and the concentration of Cu (2% to 4%). All of such high-adiabat and relatively-lower velocity implosions are robust and repeatable. Detailed descriptions of experimental parameter scans and their comparisons to VERITAS modeling will be reported elsewhere.” In addition, the word “highly” has been removed in the main text on page 15.

With these detailed responses to Reviewer #3's points and suitable changes made, we hope the further-revised manuscript can now be accepted for publication in *Nature Communications*.

REVIEWERS' COMMENTS

Reviewer #1 (Remarks to the Author):

Dear Authors and Editor,

the article has now become clearer. I accept the answer to my last comment and the argument that "the experimental observations of enhanced broadening reported here shall motivate future investigations on how density effects change line broadening in warm-dense plasmas".

I think the article is suitable for publication.

Response to Reviewers Comments on NCOMMS-22-15716B

Response to Comments of Reviewer #1

We thank reviewer #1 for being satisfied by our answer that “*the experimental observations of enhanced broadening reported here shall motivate future investigations on how density effects change line broadening in warm-dense plasmas.*” We are glad to see the reviewer’s final assessment that “*the article is suitable for publication.*”